# OXE-AugE: A Large-Scale Robot Augmentation of OXE for Scaling Cross-Embodiment Policy Learning

Guanhua Ji [* 1 2]   Harsha Polavaram [* 1]   Lawrence Yunliang Chen [* 1]   Sandeep Bajamahal [1]   Zehan Ma [1]
Simeon Adebola [1]   Chenfeng Xu [1 3]   Ken Goldberg [1]

## Abstract

Large and diverse datasets are needed for training generalist robot policies that can control a variety of robot embodiments—robot arm and gripper combinations—across diverse tasks and environments. As re-collecting demonstrations and retraining for each new embodiment are prohibitively costly, we study whether existing robot data can be augmented to improve transfer and generalization across embodiments. The Open X-Embodiment (OXE) dataset, which aggregates demonstrations from over 60 robot datasets, has been widely used for training generalist policies. However, it is highly imbalanced: the top four robot types account for over 85% of its real data, which risks overfitting to robot–scene combinations. We present AugE-Toolkit, a scalable robot augmentation pipeline, and OXE-AugE, a high-quality open-source dataset that augments OXE with 9 different robot embodiments. OXE-AugE provides over 4.4 million trajectories, more than triple the size of the original OXE. We conduct a systematic study of how scaling robot augmentation impacts cross-embodiment learning. Results suggest that augmenting datasets with diverse arms and grippers improves policy performance not only on the augmented robots, but also on unseen robots and even the original robots under distribution shifts. In physical experiments, fine-tuning generalist policies such as OpenVLA and $\pi_0$ on OXE-AugE improves success rates by 24–45% on unseen robot–gripper combinations across four real-world manipulation tasks. Project website: https://OXE-AugE.github.io/.

[*]Equal contribution   [1]Department of EECS, UC Berkeley [2]GRASP Laboratory, University of Pennsylvania [3]Department of CS, UT Austin. Correspondence to: Lawrence Yunliang Chen <yunliang.chen@berkeley.edu>, Chenfeng Xu <xuchenfeng@utexas.edu>.

*Proceedings of the 43rd International Conference on Machine Learning*, Seoul, South Korea. PMLR 306, 2026. Copyright 2026 by the author(s).

## 1. Introduction

Large and diverse datasets have driven recent progress in general-purpose robot learning, enabling policies to generalize across tasks, objects, and embodiments (Jang et al., 2021; Brohan et al., 2023b;a; Jiang et al., 2023; Shah et al., 2023b; Lynch et al., 2023; Shridhar et al., 2021; 2022; Reed et al., 2022; Chen et al., 2023b; Driess et al., 2023). However, collecting real-world robot demonstration data remains costly and time-consuming (Lee et al., 2021; Herzog* et al., 2023; Kalashnikov et al., 2022; Khazatsky et al., 2024; Fang et al., 2023; Shafiullah et al., 2023), limiting scale by both the number of teleoperated robots and the duration of each demonstration. While simulation offers a promising path to scale (Nasiriany et al., 2024; Maddukuri et al., 2025; Dai et al., 2024), the sim-to-real gap in manipulation dynamics and perception remains a major challenge (Peng et al., 2018; Ramos et al., 2019; Lim et al., 2022; Memmel et al., 2024; Bousmalis et al., 2018; Ho et al., 2021).

This challenge is amplified by increasing hardware diversity. New robots are regularly introduced, and policies trained on one platform often fail to transfer to others. Given the high cost of recollecting demonstrations for every new platform, it is desirable to reuse existing data across embodiments. Cross-embodiment generalization—the ability to transfer policies across different robot embodiments—is thus an important goal for scalable and practical robot learning (Yang et al., 2023; Doshi et al., 2024; Black et al., 2024).

The Open X-Embodiment (OXE) dataset (Collaboration et al., 2024) represents a major step toward this goal, aggregating demonstrations from over 60 real-world robot datasets. However, OXE is highly imbalanced: over 85% of real trajectories come from just four robots (Franka, xArm, KUKA iiwa, and Google Robot). As a result, generalist policies trained on OXE often rely on the policy implicitly learning embodiment-agnostic features, without explicit mechanisms to mitigate robot bias, and in practice they often require fine-tuning on new robots, even when they are visually or kinematically similar (Kim et al., 2024; NVIDIA et al., 2025; Black et al., 2024).

Chen et al. (Chen et al., 2024b) propose robot embodiment

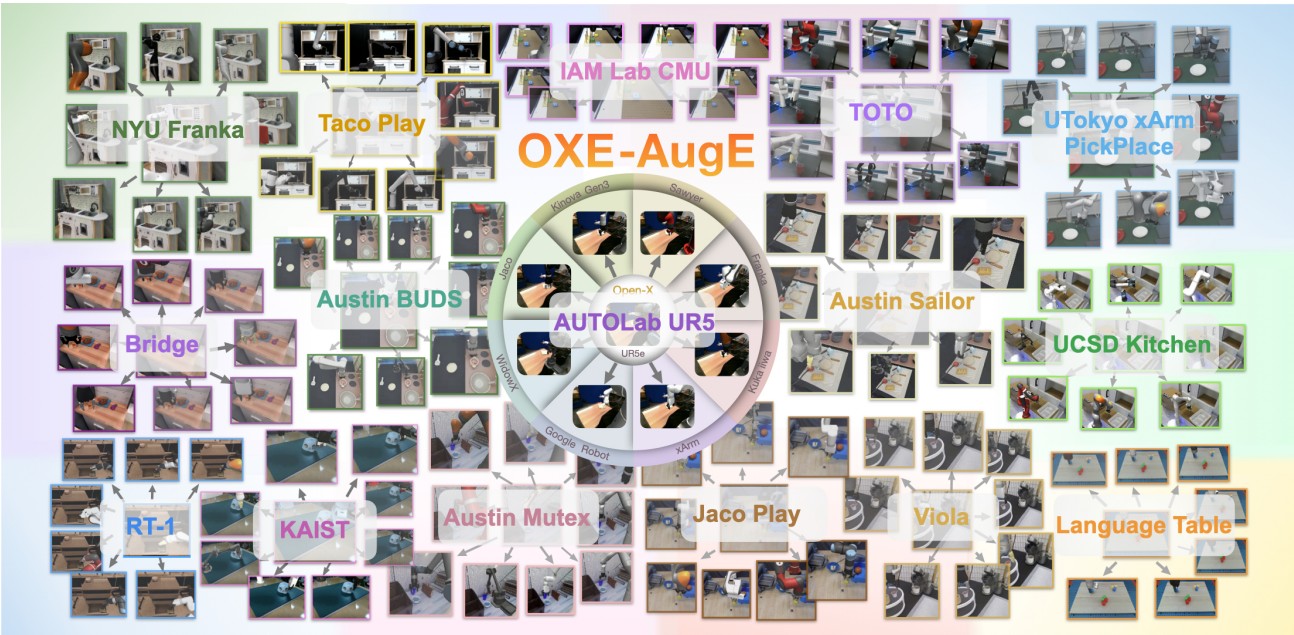

**Figure 1. We present OXE-AugE, a large-scale open-source dataset that augments the Open-X Embodiment (OXE) dataset (Collaboration et al., 2024) with 9 different robot embodiments across 16 datasets, covering 58% of the widely-used Octo pretraining mixture (Octo Model Team et al., 2024).** In total, OXE-AugE provides over 4 million trajectories, more than triple those in the original OXE. Robots in OXE-AugE include Franka, UR5e, xArm7, Google Robot, WidowX, Sawyer, Kinova3, KUKA iiwa, and Jaco. We find that training on OXE-AugE improves OpenVLA (Kim et al., 2024) and $\pi_0$ (Black et al., 2024) policy performance by up to 24-45% on previously unseen robot-gripper combinations across four real-world manipulation tasks.

augmentation—transforming demonstrations collected on one robot into synthetic versions performed by another—via *cross-painting* (Chen et al., 2024a). In this work, we significantly improve prior methods and introduce **AugE-Toolkit**, an open-source pipeline that reduces visual artifacts while improving scalability and ensuring kinematic consistency through a combination of simulation and learned models.

Using AugE-Toolkit, we generalize robot augmentation beyond pairwise transfer and study its effects at scale. We ask: (1) Does robot augmentation improve robustness on the original robot under visual perturbations? (2) Does increasing the number of augmented embodiments improve performance on augmented robots? (3) Do policies trained on diverse augmentations generalize to unseen robots? Results suggest that scaling robot augmentation leads to consistent gains, particularly for robustness and generalization. We conjecture that robot augmentation helps policies focus on task-relevant geometry rather than incidental visual features.

We present **OXE-AugE**, a high-quality open-source dataset that augments 16 popular OXE datasets with 9 different robot embodiments. The resulting dataset provides over 4 million trajectories—more than triple the size of the original OXE—and covers 58% of the widely used Octo pretraining mixture (Octo Model Team et al., 2024). By varying robot embodiment while preserving task and scene, OXE-AugE provides a new resource for training robust and transferable

visuomotor policies. We further demonstrate that generalist models such as $\pi_0$ (Black et al., 2024) and OpenVLA-OFT (Kim et al., 2025) benefit from OXE-AugE: fine-tuning on the augmented dataset improves success by 24–45% on previously unseen robot–gripper configurations across four real-world manipulation tasks.

This paper makes 4 contributions:

1. AugE-Toolkit, an improved open-source robot augmentation pipeline designed for scalable multi-robot augmentation.

2. OXE-AugE, an open-source dataset that augments 16 OXE datasets with up to 9 robot embodiments, totaling over 4.4 million trajectories and covering 58% of the Octo pretraining mixture.

3. A simulation study of how scaling robot augmentation affects generalization to both seen and unseen embodiments, and robustness to visual perturbations.

4. Physical experiments suggesting that fine-tuning foundation models on OXE-AugE can improve zero-shot success by 24–45% on novel robot embodiments.

## 2. Related Work

### 2.1. Cross-Embodiment Robot Learning

A core challenge in generalist robot learning is how to generalize across robot embodiments without collecting new

data for each platform. One common approach is domain randomization, where physical parameters of the robot (e.g., joint and link properties) are randomized in simulation to learn robot-conditioned policies (Yu et al., 2023a; Chen et al., 2018; Shao et al., 2020; Xu et al., 2021; Wang et al., 2018; Sanchez-Gonzalez et al., 2018; Pathak et al., 2019; Huang et al., 2020; Kurin et al., 2021). Hu et al. (Hu et al., 2022b) learn a world model with the robots masked out, and use visual MPC during execution time when deployed on a new robot. Another line of work explores using human data. Recent efforts have leveraged human videos (Xiong et al., 2021; Bahl et al., 2022; Duan et al., 2023; Lepert et al., 2025; Kareer et al., 2024; 2025) for robot manipulation, and motion retargeting methods (He et al., 2025; Liao et al., 2025; Allshire et al., 2025; Yin et al., 2025; Yang et al., 2025a) have been applied in locomotion settings. Other work has also explored pooling large and diverse data, including from different robots (Kalashnikov et al., 2018; Levine et al., 2018; Dasari et al., 2019; Eppner et al., 2020; Fang et al., 2023; Walke et al., 2023; Bousmalis et al., 2023) and found that the resulting policies are generalizable to new tasks and embodiments (Alayrac et al., 2022; Jang et al., 2021; Stone et al., 2023; Jiang et al., 2023; Reed et al., 2022; Radosavovic et al., 2022; Shah et al., 2023a; Yang et al., 2023; Bharadhwaj et al., 2024; Brohan et al., 2023b;a).

The Open X-Embodiment project (Collaboration et al., 2024), in particular, aggregated more than 60 datasets and demonstrated the benefit of training on various embodiments through experiments in multiple labs. Many have leveraged the OXE dataset and developed "generalist policies" that can perform multiple tasks on a wide range of robots (Octo Model Team et al., 2024; Kim et al., 2024; Black et al., 2024; Yang et al., 2024; Black et al., 2025; NVIDIA et al., 2025; Wen et al., 2025). However, the robot types in OXE are severely unbalanced, and trained policies typically perform much better on robots that are well-represented in the datasets and still require a fair amount of fine-tuning data to transfer to a new robot. Mirage (Chen et al., 2024a) proposes a test-time image inpainting pipeline to replace the new target robot in the image with the familiar source robot seen during training to achieve zero-shot cross-embodiment transfer. In this work, we inpaint the opposite direction and improve the pipeline for scalable data augmentation.

**2.2. Augmenting Real Robot Data**

Given the high cost of collecting real robot data, many approaches have explored augmenting existing datasets. Real-to-sim-to-real pipelines (Lim et al., 2022; Torne et al., 2024b;a; Li et al., 2024; Ye et al., 2025; Pfaff et al., 2025; Dan et al., 2025; Geng et al., 2025; Zhao et al., 2025; Zhang et al., 2025) reconstruct 3D object meshes from real videos and tune simulation parameters to build digital twins or "digital cousins" (Dai et al., 2024) for policy learning. Some

works (Yu et al., 2025; Yang et al., 2025c) avoid physics simulation but still require multi-view capture to build 3D Gaussian Splatting (3DGS). Many works that leverage 3DGS for rerendering and trajectory synthesis (Zhou et al., 2023a; Pan et al., 2025; Zhang et al., 2024) are primarily applicable to eye-in-hand images.

Another line of work augments data directly in image space using 2D image or video generation models, including background or object editing (Yu et al., 2023b; Chen et al., 2023c; Mandi et al., 2022; Bharadhwaj et al., 2024) and novel view synthesis (Chen et al., 2024b; Tian et al., 2024). For robot transfer, Shadow (Lepert et al., 2024) masks out robots, RoVi-Aug (Chen et al., 2024b) uses diffusion to transform robot appearance, and several works (Lepert et al., 2025; 2026; Li et al., 2025; Yang et al., 2025b; Ci et al., 2025; Xu et al., 2025) convert human videos into robot demonstrations. However, most methods focus on one-to-one transfer between a known source and target embodiment. In contrast, we study *scaling* robot augmentation across many target robots and its impact on generalization and robustness in both simulation and real-world settings.

**2.3. Study of Scaling in Robot Learning**

Several recent studies have analyzed how robot learning performance scales with data volume and model capacity (Sartor & Thompson, 2024). Models such as VIMA (Jiang et al., 2023), RT-1 (Brohan et al., 2023b), Octo (Octo Model Team et al., 2024), and HPT (Wang et al., 2024) report consistent gains when increasing dataset size and model scale. For data scaling, Lin et al. (Lin et al., 2024) show approximate power-law improvements in generalization as the number of environments and objects increases, with diversity often more important than additional demonstrations per setting, while Saxena et al. (Saxena et al., 2024) further dissect scaling effects across viewpoint, spatial layout, and object diversity. For embodiment scaling, OXE (Collaboration et al., 2024), RoboCat (Bousmalis et al., 2023), and Cross-Former (Doshi et al., 2024) find that training on multiple robot types improves transfer over single-robot training, and Yang et al. (Yang et al., 2024) demonstrate benefits from jointly training across manipulation and navigation domains. In this work, rather than pooling additional real data from multiple robots, we study scaling *robot augmentation*—synthetically generating data of multiple robot embodiments—and evaluate its effect on performance across original, augmented, and unseen robot configurations.

# 3. Problem Statement

We consider the standard imitation learning setting (Pomerleau, 1988; Ross et al., 2011), where we have a demonstration dataset $\mathcal{D}^{\mathcal{S}} = \{\tau_1^{\mathcal{S}}, \tau_2^{\mathcal{S}}, ..., \tau_n^{\mathcal{S}}\}$ consisting of $n$ successful trajectories performed by a source robot $\mathcal{S}$. Each

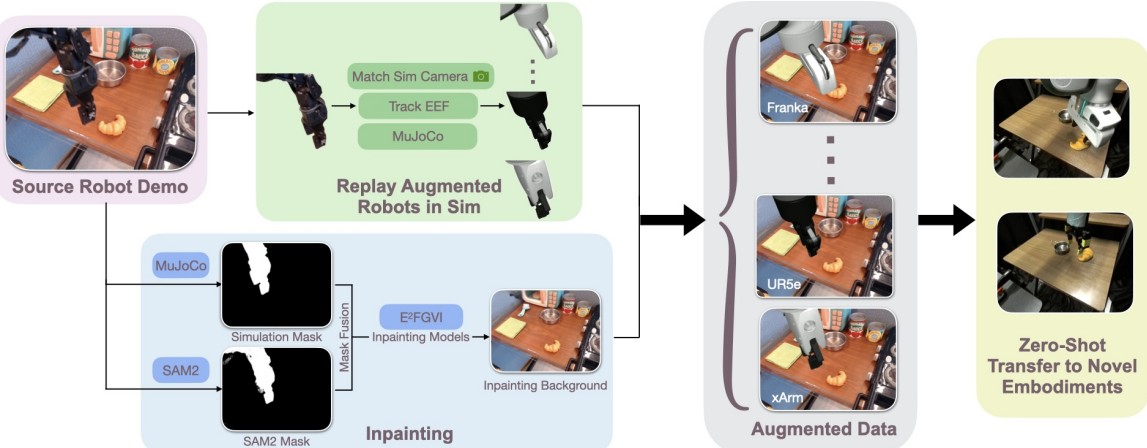

**Figure 2. AugE-Toolkit pipeline.** Given a source robot image and its corresponding robot poses, AugE-Toolkit fuses a learned SAM2 mask with a simulation-rendered mask to segment the robot, inpaints the background via E²FGVI (Li et al., 2022), and replays the same trajectory with another robot in simulation (Zakka et al., 2025). The augmented robot is composited into the reconstructed scene to form the augmented video. Policies are trained on both real and augmented data and evaluated on unseen embodiments.

trajectory $\tau_i^{\mathcal{S}} = (o_{1:H_i}^{\mathcal{S}}, p_{1:H_i}^{\mathcal{S}}, a_{1:H_i}^{\mathcal{S}})$ contains RGB observations $o_{1:H_i}^{\mathcal{S}}$, corresponding gripper 6D poses $p_t^{\mathcal{S}}$, and actions $a_t^{\mathcal{S}}$ for timesteps $t = 1, \ldots, H_i$.

We study *robot augmentation*—synthetically transforming each trajectory $\tau_i^{\mathcal{S}}$ into a corresponding trajectory $\tau_i^{\mathcal{R}}$ for a different robot embodiment $\mathcal{R}$ (arm and gripper) performing the same task. Given $\mathcal{D}^{\mathcal{S}}$ and the kinematic models of the robots (e.g., URDF), this yields a synthetic dataset $\mathcal{D}^{\mathcal{R}}$ with aligned images, poses, and actions $(o_{1:H_i}^{\mathcal{R}}, p_{1:H_i}^{\mathcal{R}}, a_{1:H_i}^{\mathcal{R}})$. Following prior work (Chen et al., 2024b), we assume the grippers across robots are similar in shape and function (e.g., 2- or 3-jaw grippers), enabling all robots to perform the same task with a shared strategy. Similar to prior work (Shah et al., 2023a; Chen et al., 2024a; Yang et al., 2023; 2024), we use Cartesian control and assume that all robots have known kinematics so the action spaces can be aligned.

Let $\mathcal{R}_{\text{Aug}} = \{\mathcal{R}_1, \mathcal{R}_2, \ldots, \mathcal{R}_N\}$ denote the set of robot embodiments used for augmentation. We train a policy on $\mathcal{D}^{\text{Train}}$, which may consist of $\mathcal{D}^{\mathcal{S}}$, $\mathcal{D}^{\text{Aug}} = \bigcup_{i=1}^{N} \mathcal{D}^{R_i}$, or both, and evaluate it on a target robot $\mathcal{T}$ at test time. We study robot augmentation's effect on (1) **robustness** ($\mathcal{T} = \mathcal{S}$), (2) **transfer** ($\mathcal{T} \in \mathcal{R}_{\text{Aug}}$), and (3) **generalization** ($\mathcal{T} \notin \mathcal{R}_{\text{Aug}}$) as $|\mathcal{R}_{\text{Aug}}|$ scales from 1 to $N$.

## 4. Methods

### 4.1. Preliminaries: Cross-Painting Framework

Cross-painting (Chen et al., 2024a;b; Lepert et al., 2025) is a three-stage pipeline applied to each image in a robot trajectory: (i) *source-robot segmentation*, (ii) *background inpainting*, and (iii) *augmented-robot replay and composit-*

*ing*. The goal is to replace the robot embodiment in each frame while preserving task and scene context.

Existing implementations differ mainly in how each stage is realized. Learning-based methods use diffusion models to modify the robot directly in pixel space (Chen et al., 2024b), requiring no explicit calibration. However, they lack kinematic guarantees and scale poorly to many target robots, since each new embodiment typically requires a separately trained model. Simulation-based methods (Chen et al., 2024a) render the robot using known camera parameters, ensuring geometric fidelity. However, these methods were originally designed for test-time adaptation, where accurate calibration is feasible, but are difficult to apply to large-scale offline datasets that often lack such information. We present AugE-Toolkit, which builds on this framework to enable scalable robot augmentation.

### 4.2. AugE-Toolkit: Scalable Robot Augmentation

AugE-Toolkit (Fig. 2) extends cross-painting in RoVi-Aug (Chen et al., 2024b) with three key components.

**(1) Fusion of Simulation and Learned Masks.** To obtain accurate robot masks without precise camera calibration, we combine simulation-provided and learned segmentation. We fine-tune SAM2 (Ravi et al., 2024) on a small labeled subset (20 trajectories from each of 16 OXE datasets). While the learned masks align well with image appearance, they often over- or under-segment near the gripper. Simulation masks are geometrically accurate but may be globally misaligned due to unknown camera poses. As such, we align, prune, and fuse the two masks. This allows us to correct for calibration errors and enables accurate rendering even on uncalibrated

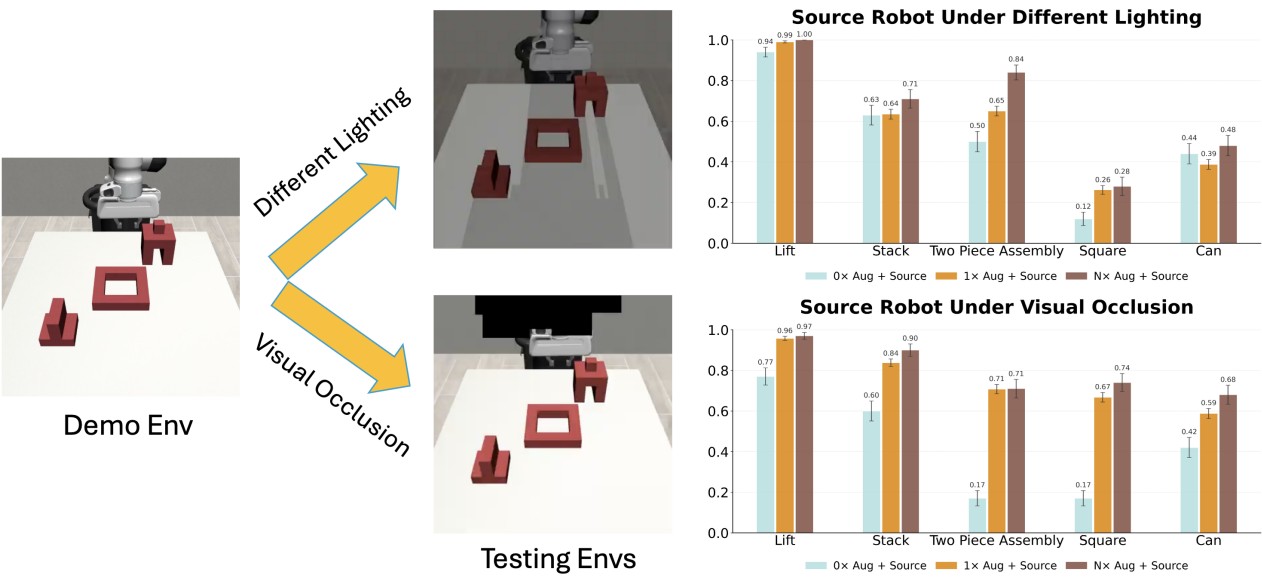

**Figure 3. Robot augmentation improves robustness on the source robot under visual perturbations. Left:** We consider two types of perturbations: different lighting conditions and visual occlusions. **Right:** Performance of policies trained on various augmented datasets on the Franka (source) robot. The performance of policies without augmentation severely degrades, while increasing the number of augmented robots makes policies more robust.

datasets; details are in the Appendix.

**(2) Automatic Base Position Selection.** To accommodate robots with different kinematic reach and scale, for each trajectory and each target robot, we algorithmically search for a base position to ensure the source trajectory can be closely tracked with operational space control (OSC) in simulation without self-collision. Starting from an initial base pose, we iteratively perform line search along the $(x, y, z)$ axes and compute tracking error, until the maximum error falls below 1 cm or a maximum iteration count is reached. Trajectories without feasible base positions are filtered out.[1] This procedure enables consistent augmentation across compact arms (e.g., WidowX) and larger robots (e.g., Google Robot) while preserving motion fidelity.[2]

**(3) Scalable Multi-Robot Deployment.** We employ simulation rendering, which reduces geometric artifacts and has better temporal coherence and pose accuracy compared to using generative models, while remaining learning-free. AugE-Toolkit supports a large collection of robot URDFs (Zakka et al., 2022). Adding a new robot only requires registering its model; no retraining or calibration is needed. Additionally, each stage of the pipeline is embar-

rassingly parallel, making it efficient to perform large-scale multi-robot augmentation. See Appendix for details.

## 5. Scaling Robot Augmentation: A Systematic Study in Simulation

We begin with a systematic simulation study to examine how robot augmentation scales along axes of transfer, generalization, and robustness. While prior work (Chen et al., 2024b) has shown that augmenting demonstrations from a source robot to a known target enables zero-shot transfer, our goal is to investigate whether robot augmentation provides broader benefits when scaled up.

We follow the Mirage evaluation setup (Chen et al., 2024a) and consider five Robosuite (Zhu et al., 2020) tasks: LIFT, STACK, CAN, TWO PIECE ASSEMBLY, and SQUARE. For each task, we use source demonstrations collected on a Franka robot. We then augment the demonstrations into $N = 4$ additional robots: UR5e, Kinova Gen3, Sawyer, and Jaco (which has a 3-jaw gripper). We use Diffusion Policy (Chi et al., 2023) to train separate policies for each condition using RoboMimic (Mandlekar et al., 2021); see Appendix for training details.

**Training configurations.** We consider four data regimes:

- **No Augmentation ("0× Aug + Source"):** Train only on the original Franka demonstrations ($\mathcal{D}^{\text{Train}} = \mathcal{D}^{\mathcal{S}}$).

- **1 Robot Augmentation ("1× Aug"):** Augment the source data into a single target robot $\mathcal{R}$ and train only on the augmented demonstrations ($\mathcal{D}^{\text{Train}} = \mathcal{D}^{\mathcal{R}}$). This

---

[1]In practice, two datasets (RT-1 Fractal and Language Table) have many trajectories that are unreachable by the WidowX robot; we exclude these from their augmentations. All other datasets are augmented into all 9 robot-gripper combinations, with >95% of trajectories achieving replay errors under 0.25 cm. See Appendix for details.

[2]For mobile robots, the optimized base translation can be interpreted as a movement action.

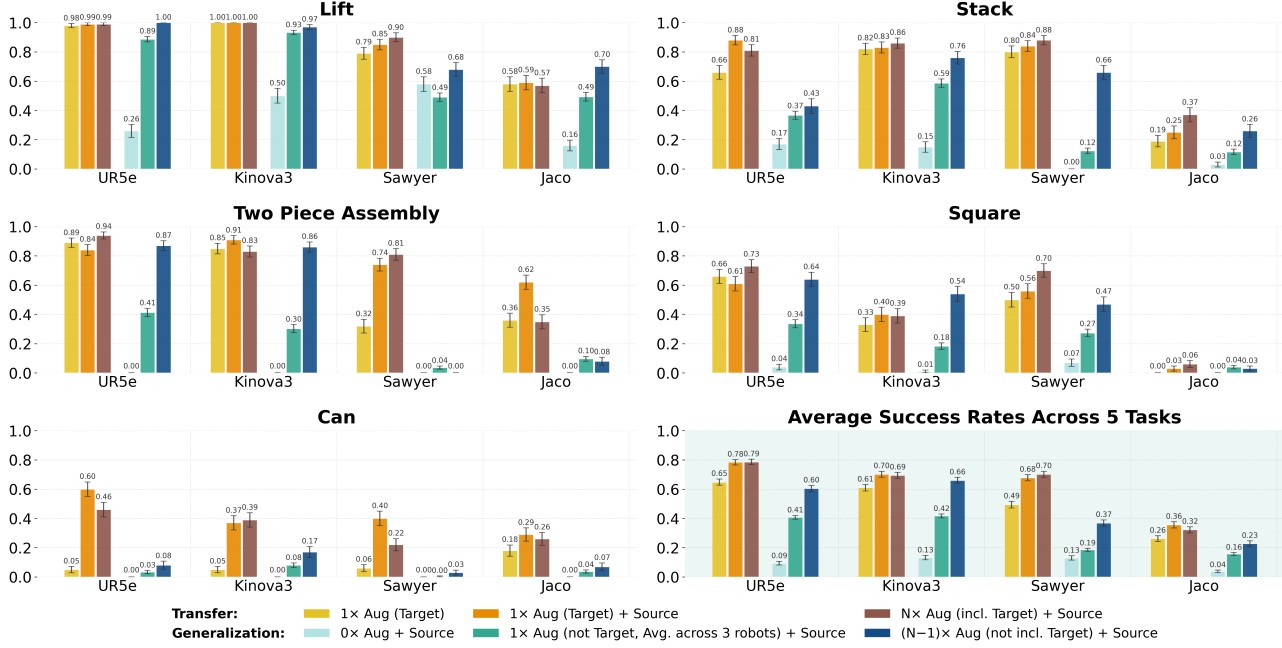

**Figure 4. Simulation experiments on scaling robot augmentation.** We evaluate how scaling the number of augmented robots affects (1) **Transfer**: performance on augmented robots (orange), and (2) **Generalization**: performance on unseen robots (blue). For transfer, we compare policies trained on the augmented target robot only ("1× Aug (Target)"), on source data plus the target robot ("1× Aug (Target) + Source"), and on source data plus all $N$ augmented robots ("$N$× Aug (incl. Target) + Source"). For generalization, we compare training on source data only ("0× Aug + Source"), on source data plus one non-target robot ("1× Aug (not Target) + Source"), and on source data plus $N-1$ robots, holding the target out ("($N$-1)× Aug (not incl. Target) + Source"). In transfer, "1× Aug + Source" substantially outperforms "1× Aug," and "$N$× Aug + Source" achieves comparable or slightly better performance. In generalization, performance improves consistently with augmentation diversity, with "($N$-1)× Aug (not incl. Target) + Source" often rivaling "1× Aug (Target)." Error bars indicate standard error over evaluation trials, aggregated across trials and tasks.

is the standard setting studied in prior work.

- **1 Robot Augmentation Together with Source Data ("1× Aug + Source"):** Combine source and one target robot's data ($\mathcal{D}^{\text{Train}} = \mathcal{D}^{\mathcal{S}} \cup \mathcal{D}^{\mathcal{R}}$).
- **Multi-Robot ("$N$× Aug + Source"):** Combine source data with all $N$ augmented robots ($\mathcal{D}^{\text{Train}} = \mathcal{D}^{\mathcal{S}} \cup \bigcup_{i=1}^{N} \mathcal{D}^{\mathcal{R}i}$, $\mathcal{R}_i \in \{$UR5e, Kinova Gen3, Sawyer, Jaco$\}$).

All data compositions are trained for the same number of steps and to convergence, ensuring equal data volume.

**Evaluation protocols.** Following Sec. 3, we evaluate each policy under three conditions:

- **Robustness (Source robot):** Evaluate on the original source robot ($\mathcal{T} = \mathcal{S}$) under visual perturbations (lighting shifts and occlusions). We compare policies trained on "0× Aug + Source," "1× Aug + Source," (averaged across 4 augmented robots) and "$N$× Aug + Source."
- **Transfer (Augmented robots):** Evaluate on robots used for augmentation ($\mathcal{T} \in \mathcal{R}_{\text{Aug}}$). We compare "1× Aug (Target)," "1× Aug (Target) + Source," and "$N$× Aug (incl. Target) + Source" settings.

- **Generalization (Unseen robots):** Evaluate on robots excluded from the training augmentations. We compare "0× Aug + Source," "1× Aug (not Target) + Source," (averaged across 3 augmented robots) and "($N$-1)× Aug (not incl. Target) + Source," where "$N$−1" excludes one robot from training and evaluates on it.

### 5.1. Study Findings

Figs. 3 and 4 summarize average success rates (± s.e.) across five tasks, each evaluated with 100 trials.

**Robustness.** Fig. 3 shows performance under test-time visual perturbations to the source robot: lighting shifts that introduce shadows and occlusions from randomly placed black patches. Policies trained only on the source robot degrade substantially, while augmentation with one or more robots markedly improves robustness; using $N$ augmented robots consistently achieves the highest success. This suggests that scaling robot augmentation enhances robustness by encouraging the policy to focus on task-relevant structure, such as the spatial relationship between gripper and object, rather than incidental features such as arm texture or lighting cues.

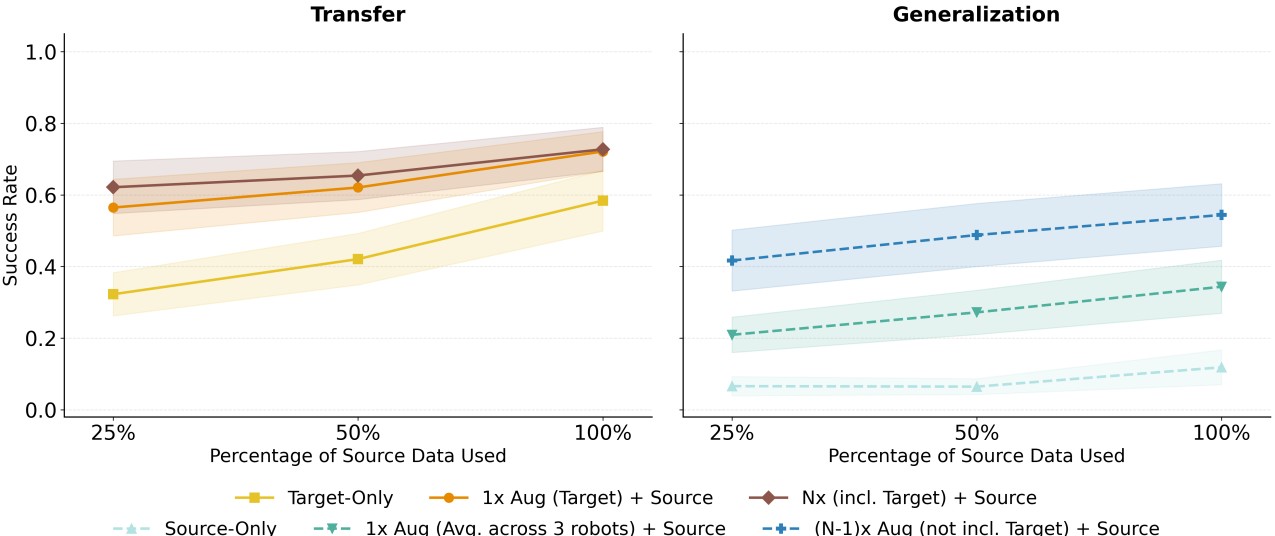

**Figure 5. Scaling with source trajectory data size and robot augmentation.** Average success rates across 5 simulation tasks for policies trained with different fractions of source trajectory data and different numbers of augmented robot embodiments. Robot augmentation improves performance at every data scale. In transfer, adding more augmented robots provides the largest gains when trajectory data is limited. In generalization, increasing embodiment diversity consistently improves performance and can be more beneficial than increasing trajectory count alone. Each policy is evaluated with 100 trials. Error bands indicate one standard error estimated via bootstrapping.

**Transfer.** On augmented robots, training only on the target augmentation ("1× Aug (Target)" ) achieves 26–65% success, while combining real and augmented data ("1× Aug (Target) + Source") improves performance by 9–19%. Training with all $N$ augmented robots performs best overall, though gains over single-robot augmentation are modest. This suggests that, for robots already seen during augmentation, much of the benefit comes from exposure to the correct embodiment, while additional robot diversity provides smaller but generally positive gains.

**Generalization.** On unseen robots, policies without augmentation perform poorly. While single-robot augmentation generalizes moderately, $(N-1)\times$ augmentation yields substantially higher success across all targets and often rivals policies trained directly on the augmented target robot ("1× Aug (Target)").

Overall, these results suggest that robot augmentation is not only useful for pairwise transfer to a known target robot, but also beneficial as a general training strategy. Scaling the number of augmented embodiments improves robustness, supports transfer to augmented robots, and enables stronger generalization to unseen robots by encouraging policies to rely less on any single robot's appearance. Appendix A.5 provides a quantitative scaling analysis of generalization performance as embodiment diversity increases.

### 5.2. Effect of Source Trajectory Data Size

We study how the amount of source trajectory data affects the effectiveness of robot augmentation. Specifically, we subsample the source robot trajectories used for augmentation to 25% and 50% of the data used in Section 5.1, repeat the experiments from Figure 4, and average policy performance across all tasks and target robots. Because each source trajectory is augmented into the same set of robot embodiments, reducing the number of source trajectories proportionally reduces the total amount of training data.

Figure 5 shows that increasing either the number of source trajectories or the number of augmented embodiments improves performance. In the transfer setting, co-training on source data and augmented target-robot data consistently outperforms training on the augmented target robot alone across all data scales. Moreover, the gains from adding more augmented robots are largest in the low-data regime, suggesting that embodiment diversity is especially valuable when trajectory data is limited. In the generalization setting, increasing the number of augmented robots yields consistent gains across all source-data scales. Notably, policies trained with more augmented robots using only 25% of the source trajectories outperform policies trained with fewer augmented robots using 100% of the trajectories, indicating that scaling embodiment diversity can be more effective than scaling trajectory count alone.

### 5.3. Comparison with Baselines

We compare cross-painting–based robot augmentation with two baselines.

**Shadow.** We compare against Shadow (Lepert et al., 2024), a strong cross-embodiment transfer method that masks robot

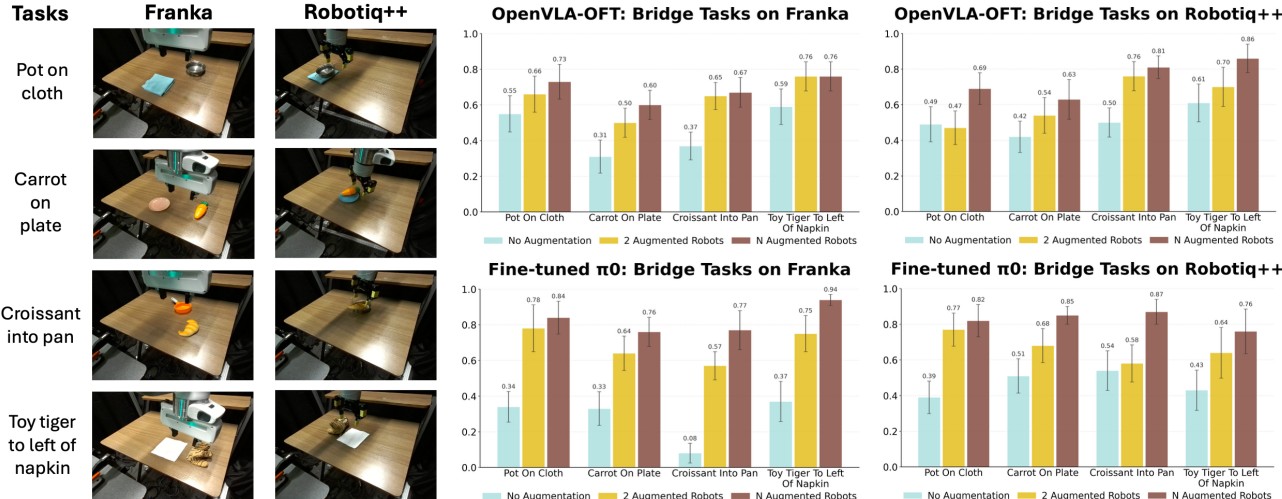

**Figure 6. Physical experiments. Left**: Illustration of the 4 tasks and the 2 testing embodiments. "Franka" is a Franka robot equipped with the default Franka gripper, and "Robotiq++" is a Franka robot equipped with a custom modified Robotiq gripper with colorful padding. **Right:** Performance of fine-tuned OpenVLA and $\pi_0$ policies trained on the Bridge subset of OXE-AugE and its $N = 8$ robot augmentations. Each policy is evaluated with 10 trials per task for each embodiment, which amounts to 80 trials per policy.

pixels and performs test-time image editing to reduce the train-test visual gap. We evaluate Shadow and AugE-Toolkit while scaling the number of augmented robots under transfer and generalization settings. AugE-Toolkit matches Shadow for generalization to unseen robots and outperforms it for transfer by 25% on average. Full results are provided in the Appendix.

**Wrist-view policies.** We also evaluate policies trained using only wrist-camera observations as a baseline for embodiment transfer. While wrist-view policies remove visual differences in robot arms, we find they are highly sensitive to gripper geometry and appearance, resulting in consistently low transfer success across target robots. The Appendix provides the detailed breakdown.

# 6. OXE-AugE: A Large Open-Source Robot Augmentation Dataset

Motivated by the simulation results in Section 5, we present **OXE-AugE**, a large-scale robot-augmented dataset derived from the Open X-Embodiment (OXE) collection (Collaboration et al., 2024). OXE-AugE is designed to scale the benefits of robot augmentation by applying cross-painting to a broad range of tasks, scenes, and robot embodiments.

We select 16 datasets from OXE that are commonly used in training robot foundation models (Firoozi et al., 2025; Collaboration et al., 2024; Octo Model Team et al., 2024; Kim et al., 2024; Black et al., 2024; NVIDIA et al., 2025; Black et al., 2025). Each original dataset was collected using a single robot platform—one among Franka, UR5, xArm, WidowX, Google Robot, and Jaco. We augment each dataset with up to 9 different robots: the 6 aforementioned robots, as

well as Sawyer, Kinova Gen3, and KUKA iiwa. Fig. 1 shows example visualizations of cross-painted augmentations, and a detailed list of the source and available augmented robots and grippers for each dataset is in the Appendix.

Overall, OXE-AugE contains over 4.4M trajectories—3$\times$ larger than the original OXE dataset. It spans diverse manipulation scenes and robot-task combinations, substantially increasing embodiment diversity. The AugE-Toolkit is also open-sourced to enable the community to extend augmentation to new datasets or robots.

## 6.1. Physical Experiments: Fine-tuning Generalist Policies on OXE-AugE

While Section 5 focuses on the single-task setting, in this section, we evaluate whether large-scale augmentation can also benefit pretrained foundation models.

We consider two generalist policies—OpenVLA (Kim et al., 2024) and $\pi_0$ (Black et al., 2024)—and fine-tune them using OXE-AugE. For evaluation, we use tasks from the Bridge dataset (Ebert et al., 2022; Walke et al., 2023), originally collected on a WidowX robot, and test on two embodiments: (1) a Franka arm with its default parallel-jaw gripper ("Franka"), which corresponds to one of the augmented robots in OXE-AugE, and (2) a Franka arm with a custom modified Robotiq gripper ("Robotiq++"), which features colored pads to simulate a visually novel embodiment (see Fig. 6). This setup evaluates both transfer to augmented robots and generalization to unseen robot-gripper configurations.

We evaluate 4 tasks: "Put the pot on the cloth," "Put the carrot on the plate," "Put the croissant into the pan," and

"Put the toy tiger to the left of the napkin." The first three appear in Bridge, while the last is novel and absent from both Bridge and other OXE datasets, testing generalization.

For each base model, we compare "no augmentation," "2 augmented robots," and "$N$ augmented robots." Specifically, "2 augmented robots" is the base model fine-tuned on the Franka and UR5e augmentation of OXE-AugE, and "$N$ augmented robots" is the base model fine-tuned on all augmentations. All augmented robots use their default grippers, and all policies are fine-tuned for the same number of steps for fair comparison. For OpenVLA, we follow OpenVLA-OFT (Kim et al., 2025) and perform LoRA fine-tuning. For $\pi_0$, we use full fine-tuning as we find it works best. Both models use $256{\times}256$ third-person observations conditioned on language instructions. More details are in the Appendix.

### 6.2. Results

Each policy is evaluated over 10 trials per task, resulting in 40 trials per embodiment. Results are summarized in Fig. 6. Both OpenVLA-OFT and $\pi_0$'s performances are relatively low when fine-tuned only on the original Bridge data, especially on Franka, due to the visual domain shift from the black WidowX gripper to the white Franka one. Fine-tuning on OXE-AugE significantly improves cross-embodiment performance: "2 augmented robots" improves success across all tasks, and $N\times$ augmentation yields the highest success overall. On average, $N\times$ augmentation improves performance by 24% for OpenVLA-OFT and 45% for $\pi_0$. On the novel Robotiq++ embodiment, fine-tuned policies reach 75% (OpenVLA-OFT) and 82% ($\pi_0$) average success, demonstrating strong generalization.

We further compare AugE-Toolkit with the diffusion-based RoVi-Aug (Chen et al., 2024b). Following the official RoVi-Aug codebase, we train ControlNet models (Zhang et al., 2023) to augment Bridge data from WidowX to four other robots, and compare policies fine-tuned on diffusion-augmented data with their AugE-Toolkit counterparts. As shown in Fig. 7, AugE-Toolkit outperforms RoVi-Aug by 48–77%. Additional details and qualitative comparisons of the augmentations are provided in the Appendix.

## 7. Conclusion

In this work, we generalize robot augmentation beyond pairwise transfer and develop it into a scalable data pipeline for robot learning. By improving the cross-painting process, we enable high-quality augmentation applicable to many existing datasets. Through systematic simulation and real-world experiments, we show that increasing the number and diversity of augmented embodiments improves generalization to unseen robots and robustness to visual perturbations. We introduce **OXE-AugE**, a large-scale open-source extension of the OXE dataset that augments 16 widely used datasets

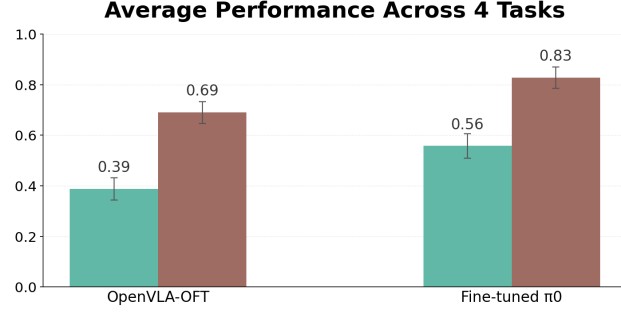

**Average Performance Across 4 Tasks**

**Figure 7. Comparison with RoVi-Aug (Chen et al., 2024b).** We compare policy performance trained on augmentation from AugE-Toolkit and RoVi-Aug. Simulation-based augmentation improves policy success by 48–77% relative to RoVi-Aug. See Appendix for a qualitative comparison.

to over 4.4M trajectories spanning 9 robot embodiments. Fine-tuning foundation models such as OpenVLA and $\pi_0$ on OXE-AugE improves success by up to 45% on previously unseen robot–gripper combinations, suggesting that explicit embodiment augmentation can complement large pooled datasets and encourage more embodiment-robust policy behavior.

**Limitations and Future Work.** AugE-Toolkit performs augmentation in 2D image space using simulation replays, which is more scalable than full 3D reconstruction but does not model accurate object-robot occlusions. It also assumes similar control strategies across robots, neglecting interaction, object collision avoidance, and dynamic differences across embodiments. Our real-world experiments primarily evaluate transfer to visually novel robot-gripper configurations; for example, Robotiq++ changes the gripper appearance but not its underlying parallel-jaw geometry. Although our simulation study covers multiple robot and gripper embodiments, we have not yet extensively evaluated real-world transfer to substantially different hardware families, humanoid hands, or complex gripper geometries. Moreover, while our current results show that robot augmentation works well in relatively simple settings involving 2- or 3-jaw grippers and third-person views, more challenging cases such as wrist-view augmentation, heavier object–gripper occlusion, substantially different camera viewpoints, and more complex manipulation tasks remain to be studied in depth. Our physical experiments focus mainly on quasi-static tasks such as pick-and-place, reflecting the task distribution of widely used OXE datasets such as Bridge and RT-1. Future work could incorporate 3D geometry and physics-aware augmentation, evaluate transfer to more geometrically diverse robots and grippers (Wu et al., 2026), extend augmentation to wrist-view, dynamic, deformable-object, or bimanual settings (Bajamahal et al., 2026), and combine embodiment augmentation with viewpoint, background, object, or task variations to further improve generalization.

## Acknowledgments

This research was performed at the AUTOLab at UC Berkeley in affiliation with the Berkeley AI Research (BAIR) Lab. L.Y. Chen was supported by the National Science Foundation (NSF) Graduate Research Fellowship Program under Grant No. 2146752. S. Adebola was supported in part by the Bakar BioEnginuity Impact Grant. The authors thank Mehdi Khfifi for some early development of our simulator, and Pannag Sanketi, Ted Xiao, Ashwin Balakrishna, and Quan Vuong for helpful discussions.

## Impact Statement

This work aims to advance large-scale, data-driven robot learning by introducing a scalable robot augmentation pipeline and the OXE-AugE dataset. By enabling high-quality, physically consistent augmentation across diverse robot embodiments, our work can accelerate research on generalist robot policies, reduce the environmental and human cost of real-world data collection, and improve reproducibility and benchmarking across labs. Potential risks include the misuse of generative augmentation methods or synthetic data to produce deceptive or misleading media, and the possibility that more capable robot policies trained on large-scale synthetic data could be applied in unsafe or unregulated contexts. Our dataset and methods are released strictly for research and educational use. We encourage future work to study mechanisms for ensuring safety, transparency, and accountability as scalable cross-embodiment robot learning continues to develop.

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

# A. Supplementary Material

## A.1. OXE-AugE Dataset Details

Table 1 presents a list of the datasets in OXE-AugE. We select 16 datasets from OXE that are commonly used in training robot foundation models (Firoozi et al., 2025; Collaboration et al., 2024; Octo Model Team et al., 2024; Kim et al., 2024; Black et al., 2024; NVIDIA et al., 2025; Black et al., 2025). The original demonstrations in those datasets were collected using Franka, UR5, xArm, WidowX, Google Robot, and Jaco platforms. A filled circle (●) indicates the source robot, and a check mark (✓) indicates robots for which augmented demonstrations are available. For 14 out of the 16 datasets, all 9 robots are available. For RT-1 (Fractal) and Language Table datasets, we find most trajectories' range of the motion exceeds the workspace of WidowX, so we exclude the WidowX augmentation. Overall, OXE-AugE contains over 550,000 demonstrations per robot, totaling 4.4M trajectories—3× larger than the original OXE dataset.

| Dataset | Franka | UR5e | xArm7 | Google | WidowX | Sawyer | Kinova3 | KUKA | Jaco |
|---|---|---|---|---|---|---|---|---|---|
| Berkeley AUTOLab UR5 (Chen et al., 2023a) | ✓ | ● | ✓ | ✓ | ✓ | ✓ | ✓ | ✓ | ✓ |
| TACO Play (Zhou et al., 2023b) | ● | ✓ | ✓ | ✓ | ✓ | ✓ | ✓ | ✓ | ✓ |
| Austin BUDS (Zhu et al., 2022b) | ● | ✓ | ✓ | ✓ | ✓ | ✓ | ✓ | ✓ | ✓ |
| Austin Mutex (Shah et al., 2023c) | ● | ✓ | ✓ | ✓ | ✓ | ✓ | ✓ | ✓ | ✓ |
| Austin Sailor (Nasiriany et al., 2022) | ● | ✓ | ✓ | ✓ | ✓ | ✓ | ✓ | ✓ | ✓ |
| CMU Franka Pick-Insert (Saxena et al., 2023) | ● | ✓ | ✓ | ✓ | ✓ | ✓ | ✓ | ✓ | ✓ |
| KAIST Nonprehensile (Kim et al., 2023) | ● | ✓ | ✓ | ✓ | ✓ | ✓ | ✓ | ✓ | ✓ |
| NYU Franka Play (Cui et al., 2022) | ● | ✓ | ✓ | ✓ | ✓ | ✓ | ✓ | ✓ | ✓ |
| TOTO (Zhou et al., 2023b) | ● | ✓ | ✓ | ✓ | ✓ | ✓ | ✓ | ✓ | ✓ |
| UTokyo xArm PickPlace (Matsushima et al., 2023) | ✓ | ✓ | ● | ✓ | ✓ | ✓ | ✓ | ✓ | ✓ |
| UCSD Kitchen (Yan et al., 2023) | ✓ | ✓ | ● | ✓ | ✓ | ✓ | ✓ | ✓ | ✓ |
| Austin VIOLA (Zhu et al., 2022a) | ● | ✓ | ✓ | ✓ | ✓ | ✓ | ✓ | ✓ | ✓ |
| Bridge (Walke et al., 2023) | ✓ | ✓ | ✓ | ✓ | ● | ✓ | ✓ | ✓ | ✓ |
| RT-1 Robot Action (Brohan et al., 2023b) | ✓ | ✓ | ✓ | ● | | ✓ | ✓ | ✓ | ✓ |
| Jaco Play (Dass et al., 2023) | ✓ | ✓ | ✓ | ✓ | ✓ | ✓ | ✓ | ✓ | ● |
| Language Table (Lynch et al., 2023) | ✓ | ✓ | ● | ✓ | | ✓ | ✓ | ✓ | ✓ |

**Table 1. Source and augmented robots in the OXE-AugE dataset.** A filled circle (●) indicates the **source robot**, and a check mark (✓) indicates robots for which augmented demonstrations are available.

In both the original OXE dataset and OXE-AugE, the Franka robot uses the default Franka Hand, UR5e uses the Robotiq 2f-85 gripper, xArm7 uses UFACTORY xArm Gripper G2, Google Robot uses the custom 2-finger gripper, Jaco uses the Kinova KG-3 gripper, and WidowX 250 uses the default parallel-jaw gripper. Additionally, OXE-AugE uses the Robotiq 2f-85 gripper for KUKA iiwa and Kinova3, and the Rethink Gripper for the Sawyer robot. We use these grippers as they are the most common types used for each robot, but switching grippers is also easy to do.

Figure 8 shows example images in OXE-AugE. Figure 9 illustrates the data sources of OXE-AugE and its relationship to OXE and the Octo training mixture. OXE v1.0 contains about 1.4M trajectories. V1.1 expands the total number of trajectories to 2.4M, however, only 1.4M of them are real robot data with manipulation or mobile manipulation (the remaining are sim, navigation, locomotion, human, or VQA data). Starting from there, we select 14 datasets out of the 25 datasets in the Octo Training Mix, collectively accounting for 58% of the total mixture weight. We also include 2 other high-quality datasets (UTokyo xArm PickPlace and KAIST Nonprehensile Objects) that are not used in Octo. Together, these 16 datasets include 0.55M trajectories and form the source of OXE-AugE. After augmenting each dataset with 8 or 9 robots, the total OXE-AugE consists of 4.4M trajectories.

## A.2. Simulation Tasks and Robots Visualization

Fig. 10 illustrates the simulation tasks and robots. We use the Robosuite environment with 5 tasks: Lift, Stack, Two Piece Assembly, Square Peg Insertion tasks, and Can Pick-and-Place. The demonstration data is performed on a Franka robot, and we evaluate on 4 target robots: UR5e, Kinova Gen3, Sawyer, and Jaco. Jaco has a 3-jaw gripper while the other robots have 2-jaw grippers.

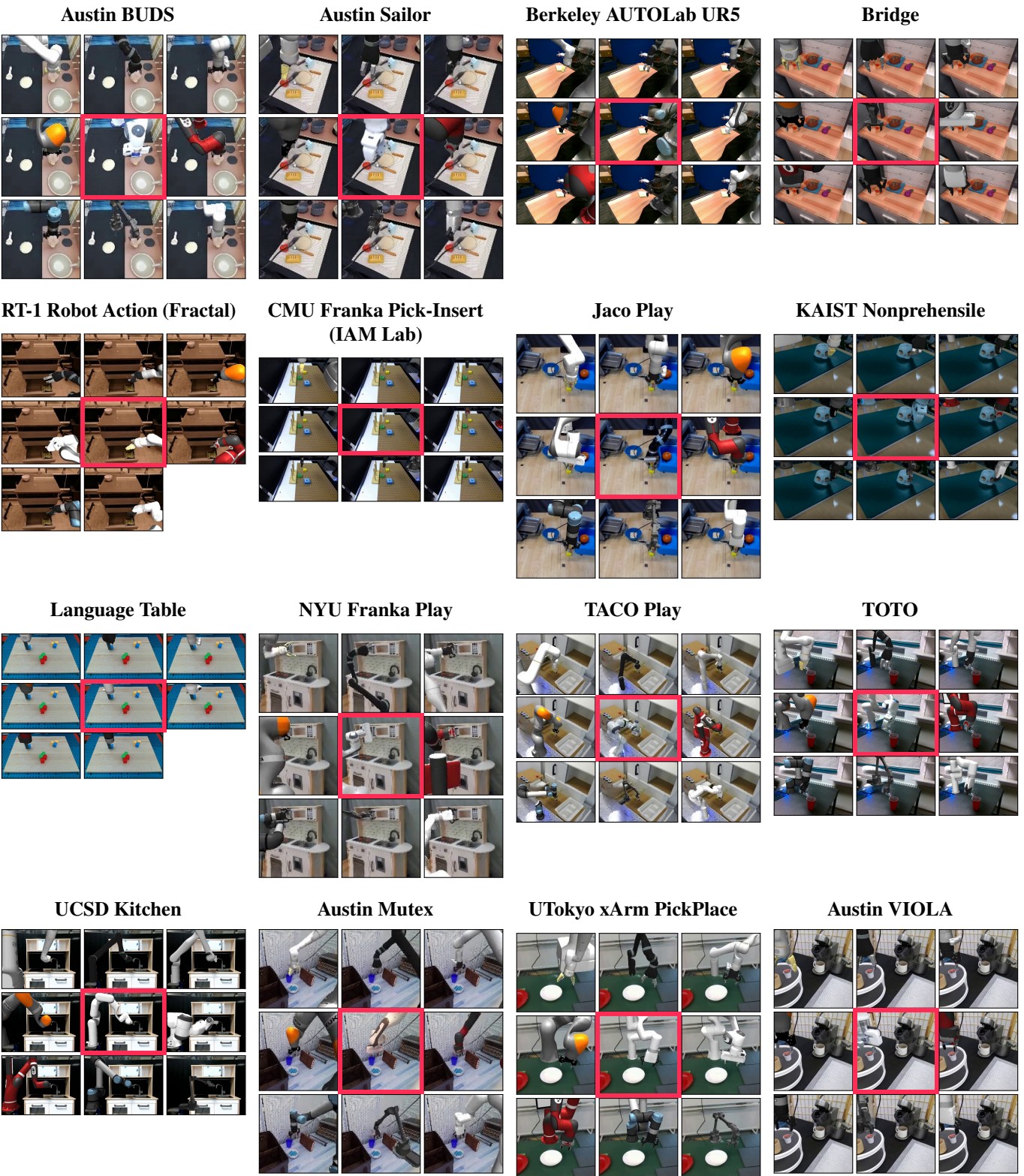

**Figure 8. Example images in OXE-AugE.** ▮ = Source robot (center cell, highlighted)

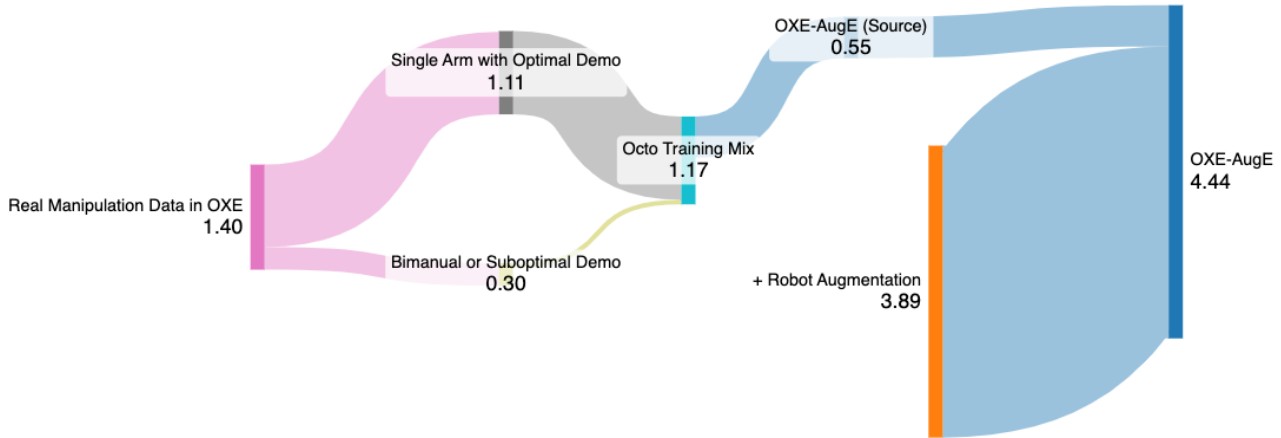

**Figure 9. Sankey diagram illustrating the data sources of OXE-AugE (numbers in millions of trajectories).** OXE v1.1 contains about 1.4M real robot manipulation trajectories. Starting from there, we select 14 datasets out of the 25 datasets in the Octo Training Mix, collectively accounting for 58% of the total mixture weight. We also include 2 other high-quality datasets (UTokyo xArm PickPlace and KAIST Nonprehensile Objects) that are not used in Octo. This set of 0.55M trajectories forms the source datasets for OXE-AugE. After robot augmentation, the total OXE-AugE consists of 4.44M trajectories.

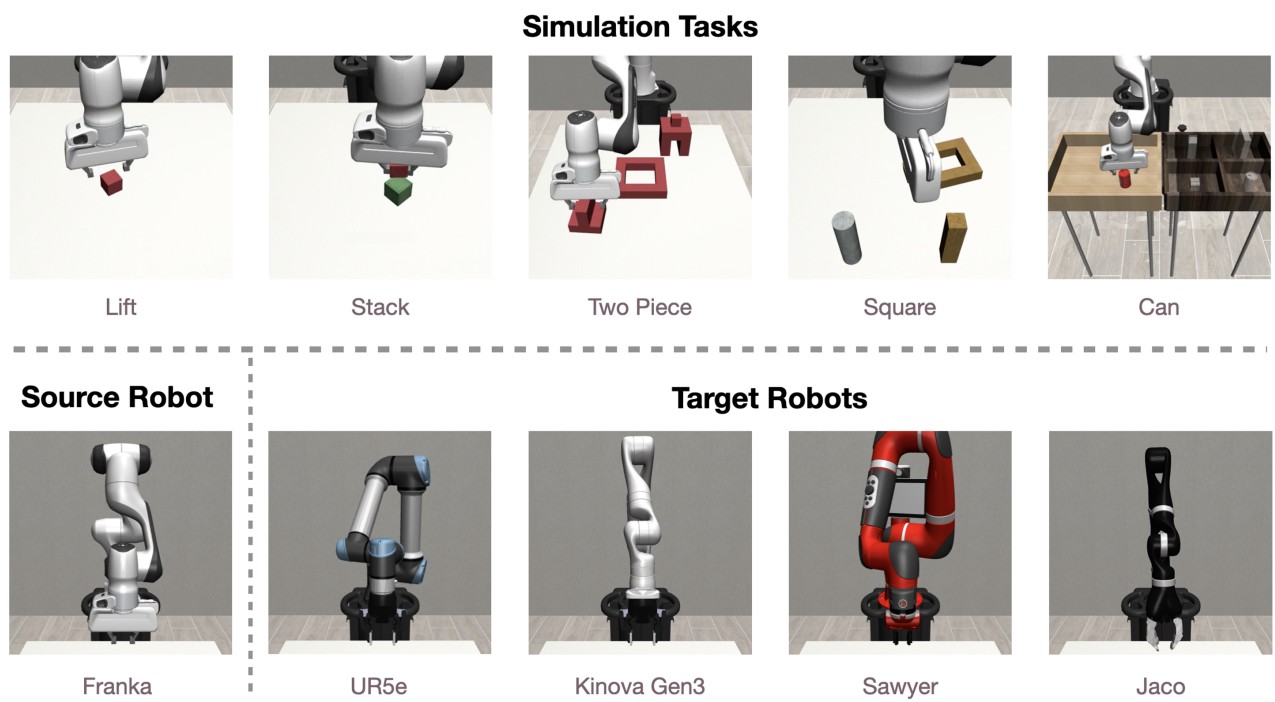

**Figure 10. Simulation tasks and robots.** We use the Robosuite environment with 5 tasks: Lift, Stack, Two Piece Assembly, Square Peg Insertion tasks, and Can Pick-and-Place. The demonstration data is performed on a Franka robot, and we evaluate on 4 target robots: UR5e, Kinova Gen3, Sawyer, and Jaco. Jaco has a 3-jaw gripper while the other robots have parallel-jaw grippers.

## A.3. AugE-Toolkit Details

### A.3.1. DETAILS ON SIMULATION AND LEARNED MASK FUSING

We align and fuse learned (SAM2) and simulated masks in three steps: (1) *Translation alignment*: shift the simulation mask with a grid search to maximize IoU with the learned mask, and the offset found is applied to the rendering of augmented robots. In addition to translational alignment, we discretely search over a small set of camera field-of-view (FoV) values during mask fusion and select the FoV that maximizes overlap between simulated and learned masks, allowing us to correct

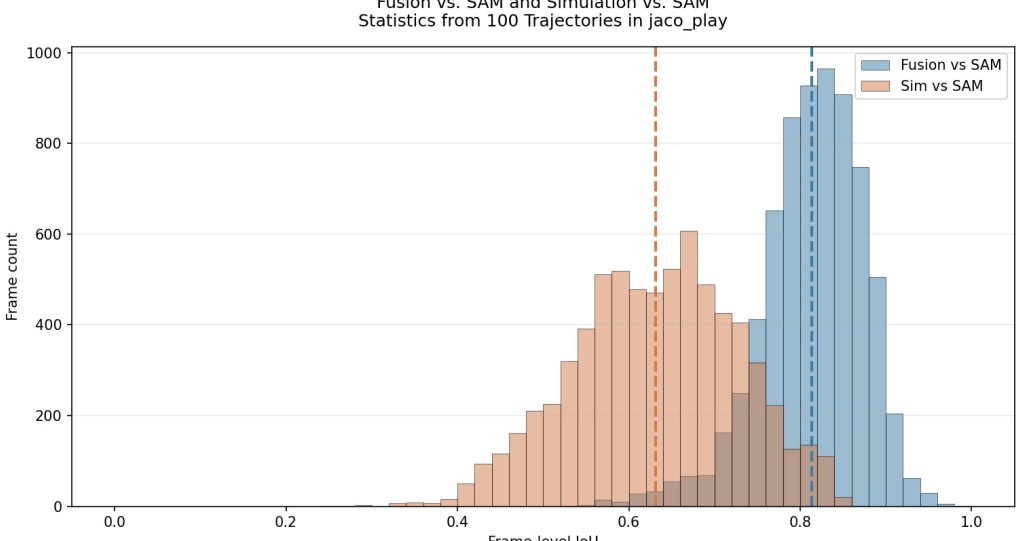

**Figure 11. IoU Distributions before and after mask fusion.** We plot the IoU between the simulation and the learned SAM2 masks before and after the mask alignment and fusion process. We see that the fusion process significantly increases the IoU (mean±std is 0.63±0.10 before fusion and 0.81±0.06 after fusion), suggesting it is effective in reducing calibration, modeling, and masking errors from simulation.

moderate intrinsic mismatches. This search is limited to a narrow FoV range and is intended to correct coarse intrinsic mismatches rather than perform full camera calibration. (2) *Pruning*: remove learned-mask pixels farther than a threshold from the aligned simulation boundary; (3) *Union and smoothing*: combine both masks and apply morphological closing. This fusion process corrects for calibration errors and enables accurate rendering even on uncalibrated datasets.

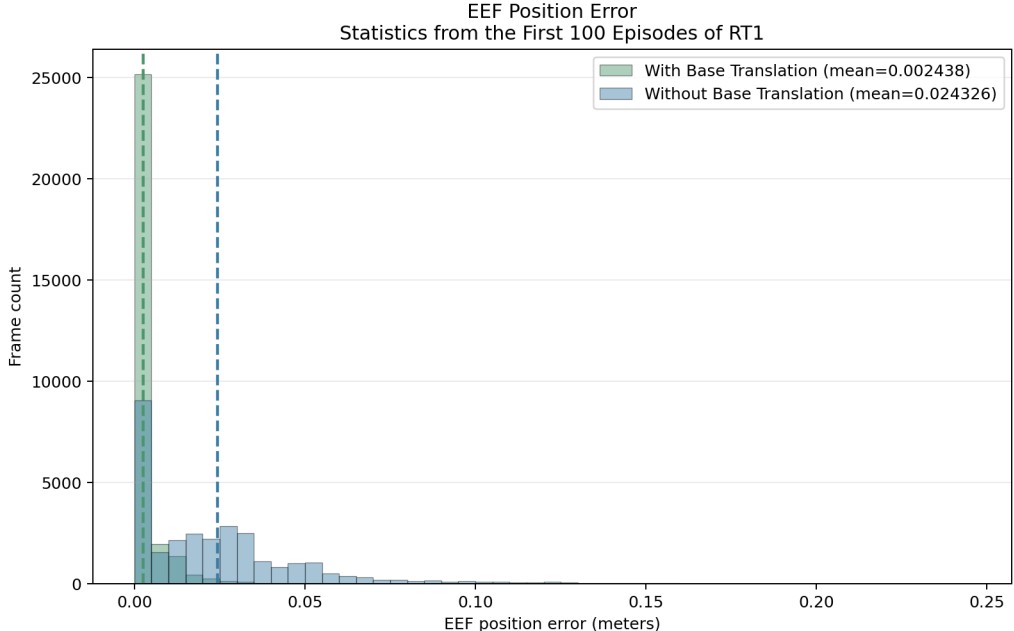

**Figure 12. Histogram of EEF tracking error before and after robot base position auto-tuning.** We visualize the distribution of target-robot tracking errors with and without the base-position selection step, using the RT-1 Robot dataset as an example. The mean and standard deviation decrease from 0.024±0.027 to 0.002±0.007 m, with the p99 error decreasing from 13 cm to 3 cm. This suggests that base-position adjustment is necessary and effective for robot augmentation.

To analyze the robustness of the mask fusion procedure, we plot the IoU between the simulation and the learned SAM2

mask before and after this procedure on random samples of OXE-AugE in Figure 11. We note that the numbers are not very close to 1 because both SAM2 masks and simulation masks are imperfect (for example, SAM2 masks may over- or under-segment, while the simulation mask may be slightly different from the real robot). However, from this plot, we can see that after the alignment step, most of the frames have >0.7 IoU alignment and empirically we also find that >0.7 IoU aligns well visually. All of the metadata are included in the augmented trajectories in OXE-AugE, and downstream training can also flag and filter data whose final masks' IoU is small (e.g., <0.6).

We further evaluate automatic base-position selection by comparing target-robot tracking errors with and without this step in Figure 12, using the RT-1 Robot dataset as an example. Without base-position selection, the augmented robot has an average tracking error of 2.4 cm at each timestep, indicating that many target poses are difficult or impossible for the augmented robot to reach. This can lead to large inpainting artifacts in frames where the augmented robot does not appear at the intended pose. In contrast, the mean tracking error is reduced to 0.2 cm after base-position selection.

### A.3.2. IMPLEMENTATION

AugE-Toolkit is implemented on MuJoCo Playground (Zakka et al., 2025), which supports a large collection of robots (Zakka et al., 2022). Each stage of the pipeline is embarrassingly parallel. Since each target robot independently replays the same trajectory, augmentation across multiple robots can be performed concurrently. A 50-frame $640 \times 480$ clip completes in approximately 25 seconds per robot; with 32-way parallelism, throughput reaches up to 75 clips per minute.

### A.4. Policy Training Details

For simulation experiments, we train diffusion policies (Chi et al., 2023) using RoboMimic (Mandlekar et al., 2021). Each policy is trained on 200 demonstrations for a single task from scratch. The demonstrations are provided by RoboMimic (Mandlekar et al., 2021) for LIFT, CAN, and SQUARE) tasks (200 each) and by MimicGen (Mandlekar et al., 2023) for STACK and TWOPIECE ASSEMBLY tasks (1,000 each), all collected on a Franka robot. The policy architecture consists of a non-pretrained ResNet18 (He et al., 2016) vision encoder and a 1D convolutional neural network (CNN) action denoiser, connected through FiLM (Perez et al., 2017). All policies are trained with a learning rate of 1e-4, batch size of 16, and for 250k steps. The visual inputs are 84x84, with random crop data augmentation during training.

For Figure 5, we exclude the Jaco robot due to the computational cost of training all combinations of source-data sizes, robot-augmentation scales, simulation tasks, and target robots. This leaves three target robots, in addition to the Franka source robot. Each data point, except for "1× Aug + Source" in the generalization setting, averages success rates over 5 tasks and 3 policies per task, with each policy evaluated over 100 rollouts. The "1× Aug + Source" generalization data points average over 5 tasks and 6 policies per task, again with 100 rollouts per policy. For LIFT, CAN, and SQUARE, the 25%, 50%, and 100% data settings correspond to 50, 100, and 200 source trajectories, respectively. For STACK and TWOPIECE ASSEMBLY, they correspond to 250, 500, and 1000 source trajectories, respectively.

For physical experiments, we fine-tune OpenVLA and $\pi_0$ using the Bridge subset of OXE-AugE and its robot augmentations. For OpenVLA, we follow OpenVLA-OFT (Kim et al., 2025) and perform LoRA fine-tuning (Hu et al., 2022a) with a learning rate of 5e-4, batch size of 8, for 25k steps. For $\pi_0$, we perform full parameter fine-tuning with a learning rate of 5e-5, batch size of 32, for 20k steps. Both models take in a third-person observation of 256×256 resolution, and are conditioned on the language instructions.

### A.5. Scaling Curves for Generalization

In this section, we quantitatively analyze how generalization performance scales with the number of robot embodiments used for augmentation. For each of the five simulation tasks, we train policies on "Source + $k \times$ Aug", where $k$ ranges from 0 to 3, and evaluate them on robots that are not included in the augmentation set. For each target robot, we average over all valid choices of the $k$ augmentation robots. For example, when $k = 2$ and the target robot is UR5e, we average over the three policies with $\mathcal{R}_{\text{Aug}} = \{\text{Kinova3, Sawyer}\}$, $\{\text{Kinova3, Jaco}\}$, and $\{\text{Sawyer, Jaco}\}$. Across all tasks and augmentation sizes, this yields 40 trained policies in total, which are evaluated on the four target robots to measure embodiment generalization.

Figure 13 shows the resulting scaling curves, following the style of (Lin et al., 2024). The $x$-axis denotes the total number of robot embodiments observed during training, including the source robot, and the $y$-axis denotes the failure rate on unseen target robots. We model the relationship between failure rate $Y$ and the number of training robots $X$ with a power law, $Y = \beta X^{\alpha}$, by fitting a linear model to the log-transformed data: $\log(Y) = \alpha \log(X) + \log(\beta)$. Dashed lines denote the

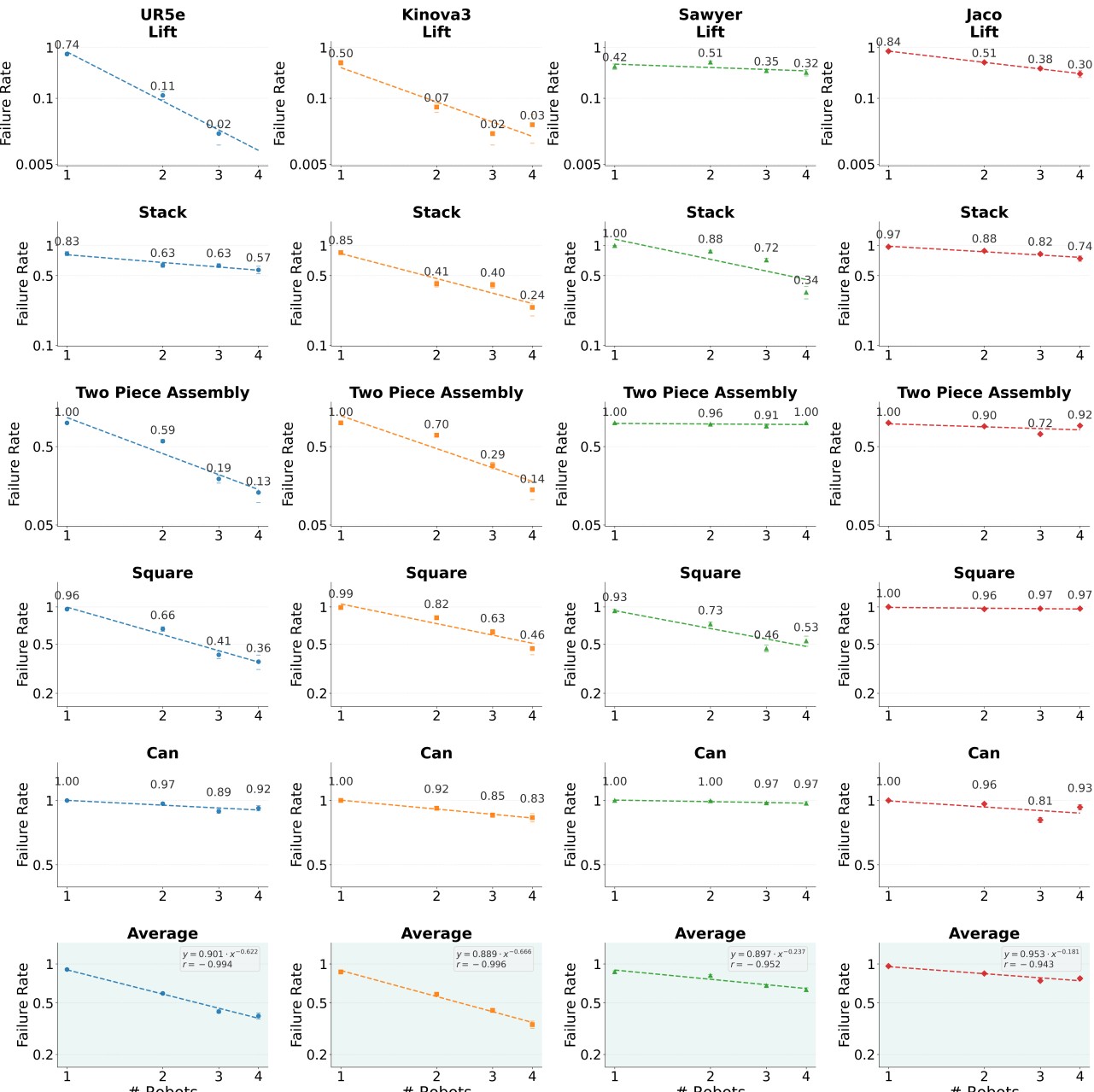

**Figure 13. Scaling curves for generalization.** Failure rate on unseen target robots as a function of the total number of robot embodiments observed during training, including the source robot. Each column corresponds to a held-out target robot, and each row corresponds to a simulation task, with the bottom row showing the average across tasks. Dashed lines indicate power-law fits of the form $Y = \beta X^{\alpha}$, where $Y$ is failure rate and $X$ is the number of training robots. The average plots report the fitted power-law coefficients and Pearson correlation coefficient $r$. Across tasks and target robots, increasing embodiment diversity generally reduces failure rates on unseen robots, with the strongest scaling trends observed for UR5e and Kinova3.

fitted scaling trends, and the average plots report the fitted power-law coefficients and Pearson correlation coefficient $r$.

Overall, failure rates decrease as more robot embodiments are observed during training, indicating that embodiment diversity improves generalization to unseen robots. While the average trends are well captured by power-law fits, the per-task curves reveal substantial task- and target-dependent variation. Some settings exhibit large reductions in failure rate as additional embodiments are added, whereas others improve more gradually or saturate. This suggests that embodiment diversity provides a consistent generalization benefit on average, but its marginal benefit depends on how well the added robots cover the relevant visual and kinematic variations of the held-out target.

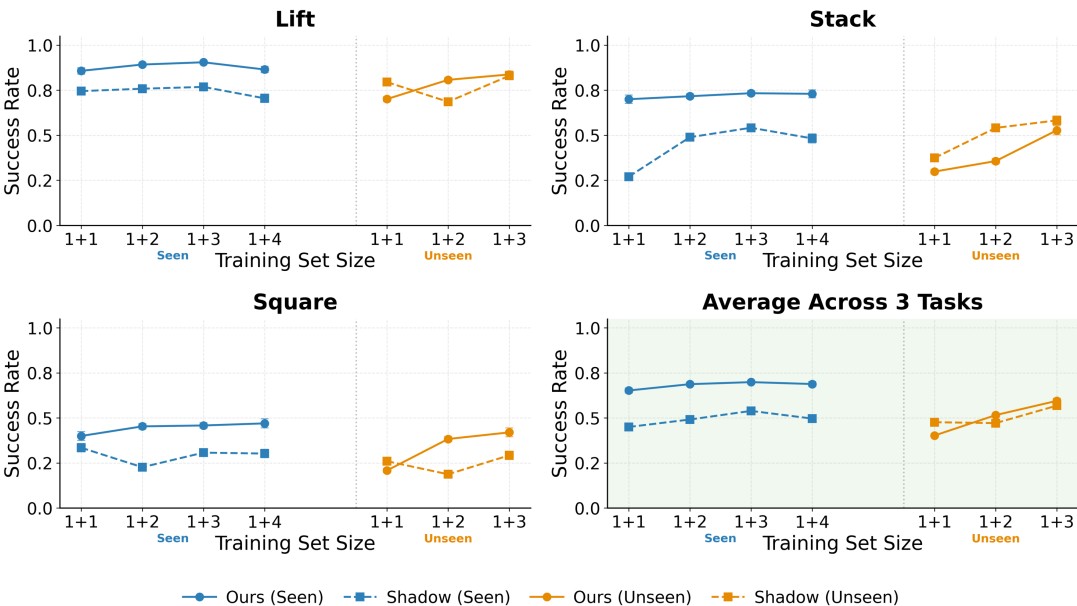

**Figure 14. Comparison with Shadow (Lepert et al., 2024).** Comparison of AugE-Toolkit and Shadow under transfer (seen robots) and generalization (unseen robots) as the number of augmented robots scales. Shadow employs real-time test-time image editing to closely match train–test visual distributions, giving it a strong advantage for cross-embodiment transfer. Despite this, AugE-Toolkit matches Shadow's performance on unseen robots and substantially outperforms it on transfer to augmented robots, particularly as embodiment diversity scales.

### A.6. Comparison with Shadow (Sim)

Shadow (Lepert et al., 2024) was originally proposed for the setting of transferring a policy trained on a single source robot to a single unseen target robot. Similar to AugE-Toolkit, Shadow uses known robot kinematics and URDF models to render robots at matched end-effector poses. However, instead of replacing the source robot with a rendered target embodiment, Shadow discards robot appearance entirely by masking robot pixels with a composite segmentation mask.

During training, Shadow replaces the source robot with a black mask corresponding to the union of the source and target robots rendered at the same pose. At test time, Shadow performs real-time image editing by masking the target robot and overlaying the source robot mask, thereby closely matching the train-time and test-time input distributions. This explicit test-time intervention substantially reduces the visual train–test gap and provides Shadow with a strong advantage for cross-embodiment transfer.

To compare Shadow with AugE-Toolkit under a scaling setting, we extend Shadow from a single target robot to $N$ augmented robots. In this setting, the policy is trained using the union of $N+1$ black robot masks. During evaluation on robots seen during training, the remaining robot masks are added to match the training-time distribution. For unseen robots, all training-time robot masks are added to approximate the same distribution.

We evaluate both methods while scaling the number of augmented robots under transfer and generalization settings; results are shown in Fig. 14. Shadow performs comparably to AugE-Toolkit for generalization to unseen robots, but underperforms significantly for transfer to augmented robots, with an average performance drop of 25%.

We attribute this behavior to the differing inductive biases of the two methods. Shadow minimizes the train–test distribution gap through test-time image editing, but at the cost of discarding all robot appearance information. As the number of robots increases, the union of black masks grows, progressively removing task-relevant visual cues and limiting the achievable performance. In contrast, AugE-Toolkit preserves physically consistent robot geometry and appearance, enabling policies to benefit from scaling embodiment diversity and achieve stronger transfer when evaluated on augmented robots.

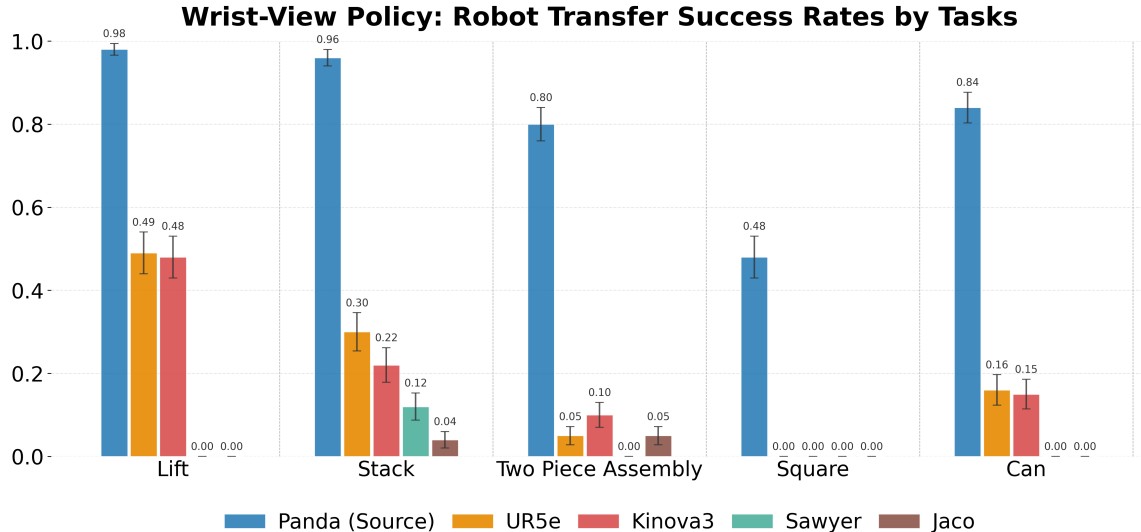

Figure 15. **Comparison with wrist-camera policies.** Transfer performance of policies trained using only wrist-camera observations across 5 tasks and 5 robot embodiments. Wrist-view policies perform well on the source robot but generalize poorly to robots with different grippers, suggesting strong sensitivity to gripper-specific visual cues and limited cross-embodiment transfer.

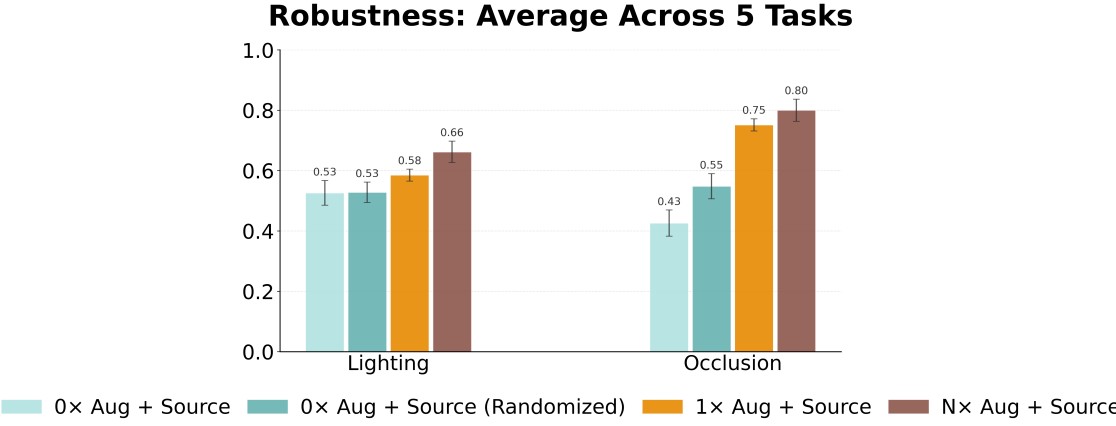

Figure 16. **Comparison with standard image augmentation on robustness.** We compare policies trained with no visual augmentation, standard visual augmentation including color jitter and noise, single-robot augmentation, and multi-robot augmentation. Results are averaged over 100 rollouts across 5 simulation tasks. While standard visual augmentation improves robustness over the no-augmentation baseline in the occlusion setting, robot augmentation significantly outperforms standard visual augmentation in both cases, especially with $N$ augmented robots.

### A.7. Comparison with Wrist-View Policies (Sim)

We evaluate policies trained using only wrist-mounted camera observations as a baseline for cross-embodiment transfer. For each task, we train a policy following the same training procedure as our method, but replace third-person observations with images from the wrist camera of the source Franka robot.

Wrist-view policies largely eliminate visual differences in robot arms across embodiments, which might suggest improved transfer. However, we find that these policies are highly sensitive to changes in gripper geometry and appearance. As shown in Fig. 15, wrist-view policies exhibit strong performance on the source robot but fail to transfer reliably to other robots, often achieving near-zero success on robots with different grippers.

Compared to the results in Fig. 4, which use third-person observations with robot augmentation, wrist-view policies achieve substantially lower transfer success across most tasks and target robots. These results indicate that removing arm appearance alone is insufficient for cross-embodiment transfer, and that wrist-view observations tightly couple policy behavior to gripper-specific visual and geometric cues, limiting generalization.

## A.8. Comparison with Standard Visual Augmentations for Robustness (Sim)

In Figure 3, we study the effect of robot augmentation on robustness under visual perturbations. For both policy-training settings, with and without robot augmentation, we use only random-crop image augmentation and exclude color jitter as well as brightness and contrast perturbations. This design isolates the effect of robot augmentation from standard image-level augmentation, showing that the observed robustness emerges from embodiment diversity rather than from direct exposure to the evaluated visual perturbations during training.

We further compare robot augmentation with standard visual augmentations, including color jitter and brightness/contrast randomization. As shown in Figure 16, averaged over 100 rollouts across Lift, Can, Square, Stack, and Two Piece Assembly, standard visual augmentation improves robustness over the no-augmentation baseline. However, robot augmentation yields a substantially larger average gain, particularly under occlusion, suggesting that embodiment-level diversity provides stronger robustness to visual distribution shifts than image-level perturbations alone.

## A.9. Applying Cross-Painting to Both Third-Person and Wrist Views (Sim)

Cross-painting is not limited to third-person views and can also be applied to wrist-camera viewpoints (Chen et al., 2024a). To evaluate this setting, we repeat the transfer and generalization experiments on three simulation tasks while augmenting both third-person and wrist-camera observations. Figure 17 shows examples of cross-painting in the wrist-camera frame.

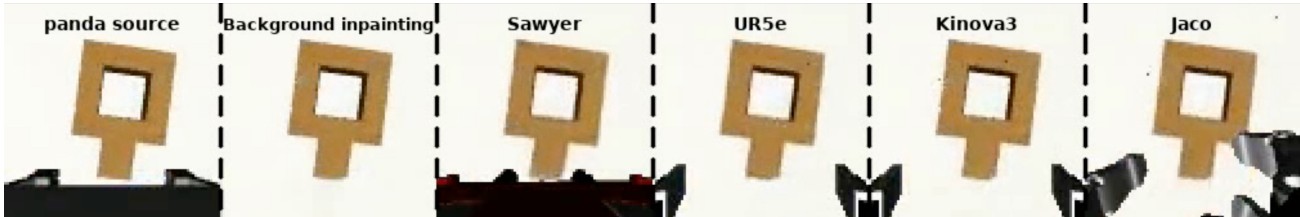

**Figure 17. Example of robot augmentation in the wrist-camera view.** Left: the source view recorded on the Franka robot. Second from left: the image after gripper masking and background inpainting. Right: cross-painted wrist-camera images with four augmented robots.

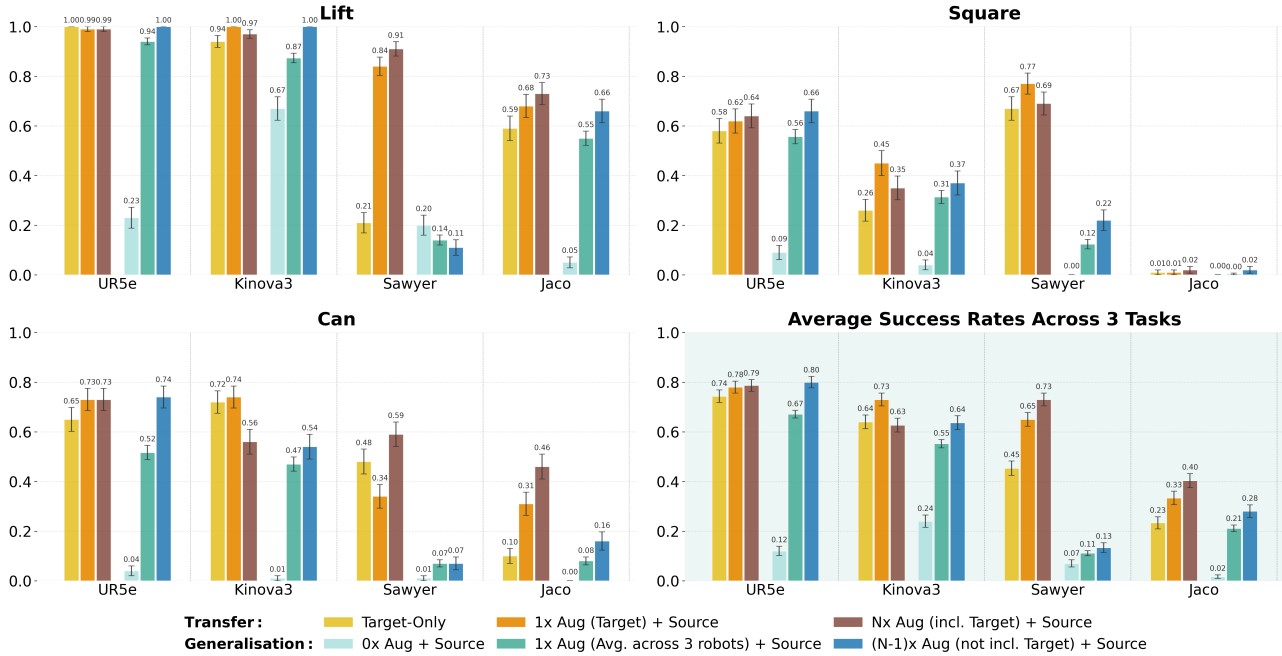

**Figure 18. Scaling robot augmentation with both third-person and wrist-camera views.** We repeat the transfer and generalization experiments from Figure 4 on three simulation tasks while augmenting both third-person and wrist-camera observations. The results show a scaling pattern similar to the third-person-only setting: using more augmented embodiments improves both transfer and generalization, with especially large gains in harder transfer settings and on unseen robot embodiments.

**Qualitative Comparisons**

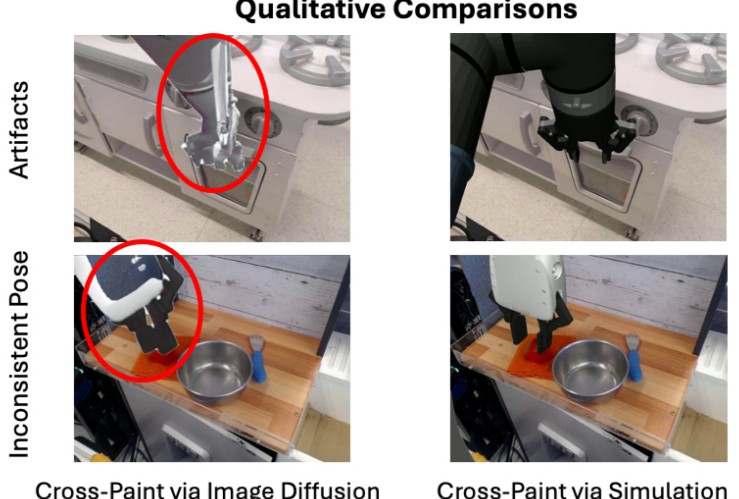

**Figure 19. Comparison with RoVi-Aug (Chen et al., 2024b).** Qualitative comparison between augmented images from RoVi-Aug (left) and AugE-Toolkit (right). The left column shows some representative artifacts produced by the trained diffusion model such as misaligned grippers and inconsistent geometry, which likely explains the inferior policy performance. The right column shows the corresponding augmentation generated by AugE-Toolkit.

Figure 18 reports the results. We observe scaling trends similar to the third-person-only setting. For transfer, training with all $N$ augmented robots achieves the best overall performance. The gains over single-target-robot augmentation are modest when the target-only baseline is already strong, but become substantial for more challenging target embodiments such as Sawyer and Jaco. For generalization, performance improves consistently as the number of augmented robots increases, and $(N-1)\times$ augmentation often approaches or exceeds policies trained directly on the target robot. These results suggest that scaling robot augmentation is also effective when both third-person and wrist-camera observations are used. However, wrist-view augmentation remains more challenging than third-person augmentation because of stronger robot self-occlusion, heavier object–gripper occlusion, and potentially larger viewpoint changes. We therefore leave a more systematic study of difficult wrist-view settings, substantially different camera viewpoints, and more complex gripper geometries to future work.

### A.10. Physical Experiment Details

In physical experiments, we use a Franka FR3 robot and a ZED 2 camera. We place the camera in roughly the same pose as that in the Bridge dataset, and crop the images to match the field of view of the Logitech camera used in Bridge. To address the challenges of different controller dynamics between the robots, we follow OpenVLA and train on the delta states of the WidowX robot instead of the action targets. At inference time, we control the Franka robot with a blocking controller, which waits for the robot to reach each commanded target state before issuing the next command. In practice, however, the achieved state may still deviate from the commanded state due to control latency and compliance. To avoid accumulating such execution errors, we feed the commanded state $s_{t+1}$, rather than the achieved state $s'_{t+1}$, back into the policy at the next step. This follows the Policy-Level Action Integrator (PLAI) proposed by Tang et al. (Tang et al., 2023).

For the 4 tasks, we use the following metric to measure policy performance: we give a score of 0.3 if the robot moves in the correct direction toward the object and attempts a grasp, a score of 0.5 if the robot successfully grasps the object, a score of 0.8 if the robot carries the object and moves toward the correct destination, and a score of 1 if the robot successfully places the object in the target position. For each task, we first uniformly randomly sample 10 paired positions for the object and target place locations from the workspace of the robot on the table (a roughly 70 cm x 50 cm region). Then for all policies and grippers, we use the same pre-selected positions for evaluation for fairness and to reduce result variance. We perform 10 trials per task, resulting in 40 trials per policy for each robot embodiment. We compute the mean and standard error of the scores for each of the 6 policies on each target embodiment in Fig. 6.

### A.11. Comparison with RoVi-Aug (Real)

To compare AugE-Toolkit with RoVi-Aug (Chen et al., 2024b), we follow its official setup and codebase. We first generate 700K paired images between WidowX and 4 different robots (UR5, Kinova, Jaco, and Sawyer) using MuJoCo (Todorov

et al., 2012) and train diffusion models (based on Stable Diffusion (Rombach et al., 2022) and ControlNet (Zhang et al., 2023)) to translate robot appearances. We then fine-tune $\pi_0$ and OpenVLA similar to Sec. A.10 but only on the original and augmented Bridge datasets. Fig. 19 shows qualitative comparisons between AugE-Toolkit with RoVi-Aug. Diffusion-based augmentations sometimes produce misaligned gripper poses or geometric inconsistencies, highlighting the benefit of simulation rendering for robot augmentation.

