# OpenReview forum: "OXE-AugE: A Large-Scale Robot Augmentation of OXE for Scaling Cross-Embodiment Policy Learning"
_ICML.cc/2026/Conference — ICML 2026 spotlight_

### Official Review · Reviewer_41Sy · 2026-02-27

**Soundness:** 4
**Presentation:** 4
**Significance:** 3
**Originality:** 3
**Overall Recommendation:** 4
**Confidence:** 5

**Summary:**

The paper introduces OXE-AugE, a large augmented dataset designed to improve cross-embodiment generalization in robot policy learning. By leveraging the AugE-Toolkit, a novel pipeline, the authors scale the original OXE dataset to over 4.4 million trajectories, encompassing 9 different robot embodiments. The AugE-Toolkit fuses learned SAM2 masks with simulation-rendered masks, enabling scalable augmentation without requiring precise camera calibration. The authors demonstrate extensively in both simulation and physical experiments that training or fine-tuning state-of-the-art Vision-Language-Action models (such as OpenVLA and $\pi_{0}$) on this dataset yields large improvements in zero-shot success rates on novel robot-gripper configurations.

**Compliance With Llm Reviewing Policy:**

Affirmed.

**Final Justification:**

I appreciate the author's rebuttal, in particular the clarification regarding the limited task scope in the current evaluation, as well as the additional ablation evidence provided for key components of the augmentation pipeline.

That said, my main concerns still remain. In particular, the current evaluation tasks are still relatively simple, and the treatment of cross-robot dynamics appears fairly simplified. In addition, the method seems primarily tailored to standard grippers, which limits its broader applicability. Taken together, these issues constrain the generality and practical scope of the paper as a robot data augmentation method.

Despite these limitations, I still find the overall methodology technically solid, and the experiments are thorough and convincing within the paper’s stated scope. Therefore, I keep my overall recommendation unchanged at 4: Weak Accept.

**Key Questions For Authors:**

No.

**Limitations:**

Yes, the authors adequately discussed the limitations and potential negative societal impact of their work.

**Strengths And Weaknesses:**

### Strength
*  **Effective Scaling and Community Contribution:** The paper proposes an effective method to scale up from single-robot data sources and successfully applies this to the OXE dataset. By substantially scaling up the dataset and releasing it open-source, this work will have a highly positive and stimulating impact on the robot learning community.

*  **Methodological Novelty:** The fusion approach combining a simulator with SAM2 is innovative. It prevents the need for precise camera calibration and is a highly effective strategy for handling the diverse, uncalibrated data sources in the OXE dataset.

* **Comprehensive Experiments:** The experimental section is thorough. The authors include both small-scale verifications to validate the effectiveness of the pipeline and large-scale experiments on the OXE dataset. Within each evaluation, the model's performance is rigorously tested across multiple configurations, providing robust and convincing evidence of the data augmentation method's effectiveness.

### Weakness

* **Lack of Task Complexity:** The real-world evaluations are restricted to relatively simple pick-and-place tasks, and the simulator tasks are similarly simple and quasi-static. These tasks are insufficient to fully demonstrate that the augmentation method is generalizable across a broader task distribution. Including more complex and demanding tasks, such as soft body manipulation or contact-rich manipulation, would significantly strengthen the evaluation.


* **Kinematic and Dynamic Discrepancies:** During the augmentation process, the authors align the end-effector (EE) poses of different robotic arms, and the authors assume the gripper is always in the foreground. However, different robotic arms have fundamentally different dynamics and kinematic structures. Consequently, the augmented data may exhibit states that are physically inconsistent with reality, potentially leading to unrealistic phenomena such as unnatural collisions or unfeasible joint configurations. While these issues did not appear in the experiments of this paper, they will inevitably occur in more general and diverse real-world manipulation tasks. Though I appreciate the authors' transparency in acknowledging this limit, it might still limit the contribution of the paper.

* **Lack of Ablation Studies:** The paper is missing necessary ablation experiments to justify the "Fusion of Simulation and Learned Masks" and the "Automatic Base Position Selection". Without these ablations, it is difficult to isolate and properly evaluate the specific contribution and necessity of each component within the overall pipeline.


* **Methodological Constraints Regarding End-Effectors:** While the authors propose a generalized data augmentation pipeline, it appears limited to standard 2-jaw or 3-jaw grippers. The paper does not mention how this pipeline would scale to or handle more complex end-effectors, such as high-DOF dexterous hands. I would appreciate it if the authors could share their perspective on this limitation.

---

> ### Author Rebuttal · Authors · 2026-03-31
>
> Thank you for your thoughtful comments and valuable feedback.
> #### **Q1: Lack of Task Complexity**
>
> As also explained in our response to Reviewer 8cbx, the scope of our physical evaluation is constrained by the task distribution covered by the available OXE data. We focused our real-world evaluation with tasks derived from those in the Bridge dataset as it is one of the largest and widely used datasets in the OXE collection. We will acknowledge in the Limitations section that evaluation on more complex with new datasets should be addressed in future work.
>
> #### **Q3: Lack of Ablation Studies: The paper is missing necessary ablation experiments to justify the "Fusion of Simulation and Learned Masks" and the "Automatic Base Position Selection".**
>
> Thanks for pointing this out. While we don’t have policy training ablation on the data without those 2 processing steps, we present the following data statistics demonstrating its value:
>
> Regarding the "Fusion of Simulation and Learned Masks" step, we compare the IoU of the two masks before and after the step, as shown in this [histogram: https://pasteboard.co/YM1ugOodEDQa.png](https://pasteboard.co/YM1ugOodEDQa.png). As also explained in our response to Reviewer VkNg, we see a big increase in IoU (from 0.63 to 0.81), which significantly reduces the amount of low-quality augmentation data one needs to filter out (IoU < 0.6).
>
> Here is the summary statistics of the distributions:
> | Metric | IoU Before Fusion | IoU After Fusion |
> |---|---:|---:|
> | Mean IoU | 0.6314 | 0.8137 |
> | Std. Dev. | 0.0962 | 0.0595 |
>
> Regarding the "Automatic Base Position Selection" step, we visualize the distribution of the target robot tracking errors with and without this base position selection in this [histogram: https://pasteboard.co/9bWg4NrF8YqR.png](https://pasteboard.co/9bWg4NrF8YqR.png), using the RT1 dataset as an example. We can see that without this step, there is an average error of 2 cm at each timestep, which suggests that there are many poses that the augmented target robot cannot reach. This would translate to a large inpainting defect in many frames where the augmented robot does not show up in the proper position.
>
> Here is the summary statistics of the distributions (unit in meter):
> | Metric | Without Base Translation | With Base Translation |
> |---|---:|---:|
> | Mean | 0.0243 | 0.0024 |
> | Std | 0.0273 | 0.0074 |
> | IQR | 0.0337 | 0.00007 |
> | P90 | 0.0543 | 0.0087 |
> | P95 | 0.0735 | 0.0136 |
> | P99 | 0.1296 | 0.0325 |
>
> We will add both discussions to the revision.
>
>
> #### **Q4: Methodological Constraints Regarding End-Effectors**
> The OXE dataset does not include much data on dexterous hands or complex grippers so we focused on 2-jaw and 3-jaw grippers. We will clarify this in the revised Limitations section and suggest it as an important topic for future work. There are some recent works that explore how one can retarget trajectories between more complex grippers (eg. Wu et al.$^*$), and studying how to scale them up would be very valuable.
>
> $^*$Wu, T., Li, S., Gong, J., Guo, C., Li, X., Mu, S., & Ding, W. (2026). CEI: A Unified Interface for Cross-Embodiment Visuomotor Policy Learning in 3D Space. IEEE Robotics and Automation Letters.

---

> > ### Author Rebuttal · Reviewer_41Sy · 2026-04-03
> >
> > I appreciate the author's rebuttal, in particular the clarification regarding the limited task scope in the current evaluation, as well as the additional ablation evidence provided for key components of the augmentation pipeline.
> >
> > That said, my main concerns still remain. In particular, the current evaluation tasks are still relatively simple, and the treatment of cross-robot dynamics appears fairly simplified. In addition, the method seems primarily tailored to standard grippers, which limits its broader applicability. Taken together, these issues constrain the generality and practical scope of the paper as a robot data augmentation method.
> >
> > Despite these limitations, I still find the overall methodology technically solid, and the experiments are thorough and convincing within the paper’s stated scope. Therefore, I keep my overall recommendation unchanged at 4: Weak Accept.

---

### Official Review · Reviewer_8cbx · 2026-03-12

**Soundness:** 4
**Presentation:** 4
**Significance:** 3
**Originality:** 2
**Overall Recommendation:** 5
**Confidence:** 5

**Summary:**

This paper presents an improvement on the OXE dataset with more robot embodiments and a better representation of different robot embodiments. Additionally the new dataset triples the size of the original dataset. The new dataset samples are produced by augmenting existing trajectories in OXE by cross-painting the original robot with a new robot embodiment (rendered in simulation). Due to different embodiment scales/workspace ranges, this paper also accounts for that by changing robot base poses accordingly to make the trajectory feasible for another robot embodiment. The paper demonstrates scaling experiments in simulation, scaling robot augmentations and testing generalization on various dimensions such as embodiments and perturbations. Real experiments also demonstrate improved success rates when training on the OXE-AugE dataset.

**Compliance With Llm Reviewing Policy:**

Affirmed.

**Final Justification:**

The paper is solid, well written, and would prove to be a valuable resource to the community. The authors have mentioned they will update their limitations section to accurately reflect the e.g pick and place nature of their tasks and lack of more complex cross embodiment tests in the real world. Never the less, neither of these are problems nor should they prevent publication since it doesn't make this resource unusable (and notably sim-to-real is more and more possible these days thanks to advances in sim technology and robot learning, so I am not concerned the tests were mostly done in simulation).

**Key Questions For Authors:**

See weaknesses

Happy to raise the score if weaknesses are addressed. I already lean towards an accept but am waiting to see other reviews and the rebuttal.

**Limitations:**

Appropriate limitations are mentioned with discussion of some interesting future work.

**Strengths And Weaknesses:**

Strengths:
- Straight forward method that demonstrates reliable data augmentation that is also fairly easy to scale up (does not need high-quality/expensive image generation).
- The experiments show very clearly that the embodiment augmentations improve cross-embodiment performance in addition to increasing same-embodiment performance in simulation. Overall the simulation experiments are well executed and very clear.
- Experiments show that image generation based approaches do not work as well as using simulation re-renders, highlighting the strength of improvements this paper makes with data augmentation.

Weaknesses:
- The real world experiments tested seem fairly simple (pick and place). Given that the OXE-AugE dataset doesn't cover some more complex tasks that models like pi0 are often tested on (e.g. clothes folding), it would have been useful to see if the cross embodiment dataset fine-tuning may enable transferrable skills to other tasks.
- One of the central claims of the paper is about generalization to unseen embodiments, however this is only tested in simulation. Is there a reason why this is not tested in real as well? It would seem appropriate to include real world testing to demonstrate cross embodiment capabilities.

---

> ### Author Rebuttal · Authors · 2026-03-31
>
> We appreciate this review and encouraging feedback. Please see our responses below as well as other threads:
>
> #### **Q1: The real-world experiments seem fairly simple (pick-and-place). Would it be useful to test whether cross-embodiment fine-tuning transfers to more complex tasks?**
>
> We will acknowledge this in the Limitations and Future Work section. This is largely because the scope of our physical evaluation is constrained by the task distribution covered by the available OXE data. Many datasets in OXE, including Bridge (Walke et al., 2023) and RT-1 Robot Action (Brohan et al., 2023b) which are among the most popular and widely used, are primarily about pick-and-place or otherwise fairly simple quasi-static tasks.
>
> We agree it would be valuable to experiment with more complex tasks. While the dataset behind $\pi_0$’s cloth folding capabilities is not open-sourced, it would indeed be useful to go beyond OXE and take other open-sourced datasets (eg. cloth folding by Yu et al.$^*$) and study cross-embodiment fine-tuning. We leave this to future work.
>
> $^*$Yu, C., Sima, C., Jiang, G., Zhang, H., Mai, H., Li, H., ... & Yuan, Y. (2026). $\chi_ {0} $: Resource-Aware Robust Manipulation via Taming Distributional Inconsistencies. arXiv preprint arXiv:2602.09021.
>
> #### **Q2: One of the central claims of the paper is about generalization to unseen embodiments, however this is only tested in simulation. Is there a reason why this is not tested in real as well?**
>
> Thank you for this important point, and we will clarify the scope of our claim. We did not evaluate on unseen physical robots because we did not have these available for experiments. However, we did try to approximate it with our Robotiq++ experiments. While “Robotiq++” is kinematically the same as other parallel-jaw grippers, its visual appearance is novel compared to all existing and augmented robots in the training dataset. In our experiments, we see that augmented policy significantly outperforms unaugmented ones, and we view this as evidence of transfer to a new robot–gripper configuration/appearance shift. Due to equipment constraints, we leave the evaluation on a completely unseen hardware family to future work, and we will clarify this limitation in the revision.

---

> > ### Author Rebuttal · Reviewer_8cbx · 2026-04-01
> >
> > Satisfied with responses. Paper is overall good and will be a great resources for roboticists

---

### Official Review · Reviewer_xhs2 · 2026-03-12

**Soundness:** 2
**Presentation:** 3
**Significance:** 3
**Originality:** 2
**Overall Recommendation:** 5
**Confidence:** 5

**Summary:**

This paper presents AugE-Toolkit, an improved cross-painting pipeline for robot data augmentation, and OXE-AugE, a 4.4M-trajectory open-source dataset that augments 16 OXE datasets with 9 robot embodiments. The cross-painting pipeline replaces the source robot in demonstration images with a target robot rendered in simulation, using three improvements over prior work (RoVi-Aug): fused simulation+SAM2 masks for accurate segmentation without camera calibration, automatic base position search for kinematic feasibility, and parallel deployment. Through systematic simulation experiments on 5 Robosuite tasks with Diffusion Policy and physical experiments on 4 Bridge tasks with OpenVLA-OFT and π₀, the authors study how scaling the number of augmented embodiments affects robustness, transfer, and generalization. The key finding is that diverse multi-robot augmentation improves generalization to unseen embodiments, and fine-tuning foundation models on OXE-AugE improves success by 24-45% on novel robot-gripper combinations.

**Compliance With Llm Reviewing Policy:**

Affirmed.

**Final Justification:**

The paper makes a solid contribution through the OXE-AugE dataset and a systematic study of how embodiment augmentation scaling affects robustness, transfer, and generalization. During the rebuttal, the authors addressed all major concerns. The additional experiments meaningfully strengthened the paper's scaling narrative. I raise my score from 4 to 5.

**Key Questions For Authors:**

1. In the robustness experiments (Section 5.1, Figure 3), the lighting shift and visual occlusion conditions are evaluated without standard data augmentation techniques such as color jitter and brightness/contrast perturbation, all of which are known to improve robustness to visual distribution shifts. Was this a deliberate choice? If the "0× Aug + Source" baseline already included these standard augmentations, how much of the robustness gap would remain? This is important for isolating the contribution of embodiment augmentation from the contribution of basic visual augmentation, which is much cheaper to implement.
2. **Suggestion: disentangling trajectory diversity from embodiment diversity.** I think the paper could benefit from a more fine-grained scaling analysis. The current evaluation compares 0×, 1×, and N× augmented embodiments, but it would be very interesting to see a 2D analysis that separately varies the number of embodiments and the number of unique trajectories per embodiment. For example, a heatmap with embodiment count on one axis and trajectory count on the other. Even a partial grid (e.g., {1, 2, 4} embodiments × {50, 100, 200} trajectories per embodiment) in simulation would help reveal whether the gains are driven primarily by embodiment diversity, trajectory volume, or their interaction. This is not a requirement, but I believe it would significantly strengthen the scaling narrative and provide more actionable guidance for practitioners deciding how to allocate their augmentation budget.
3. The "unseen" Robotiq++ embodiment (Section 6.1) differs from the training-time Robotiq only in visual appearance (colorful padding), not in gripper geometry, both are parallel-jaw grippers with the same kinematic structure. This makes it unclear whether the method enables generalization to truly novel end-effectors with different geometries (e.g., a custom UMI-style gripper). Have you tested or considered evaluating on a gripper with fundamentally different geometry? The current evaluation may overstate the method's generalization capability.
4. Cross-painting can only augment third-person views. In a dual-view setup (third-person + wrist), augmenting only the third-person stream would create inconsistent observations, as the external view shows the target robot while the wrist view still shows the source gripper. This appears to make the method fundamentally incompatible with the standard dual-view setup used in most precision manipulation deployments. (a) Have you considered this issue? (b) Do you have any evidence on how policies handle such view-inconsistent training data? (c) How do you envision extending cross-painting to wrist-camera observations, where the gripper must also be replaced to maintain consistency? This is important because it defines the practical scope of the contribution.

**Limitations:**

The authors discuss limitations in the conclusion (lines 440-458), acknowledging that augmentation is in 2D image space (not full 3D), does not model object-robot occlusions, assumes similar control strategies across robots, and neglects dynamic differences across embodiments. This is a reasonable and honest discussion. However, an important limitation is not addressed: the method is fundamentally restricted to third-person views and cannot augment wrist-camera observations, yet all experiments exclude wrist cameras and no dual-view evaluation is conducted. This is a significant practical constraint that should be discussed explicitly.

**Strengths And Weaknesses:**

**Strengths:**
- The paper addresses a genuine and important bottleneck, i.e., the severe embodiment imbalance in OXE where 4 robot types account for >85% of data. The 4.4M-trajectory open-source dataset spanning 9 embodiments and 16 datasets is a substantial community resource. The generalization finding that diverse augmentation enables zero-shot transfer to unseen embodiments without any target-specific data is a compelling and practically useful result.

- The simulation study (Section 5) is well-designed with a clean three-axis evaluation framework (robustness, transfer, generalization) and 100 trials per condition with error bars. Multi-architecture validation across Diffusion Policy, OpenVLA-OFT, and π₀ reduces the risk of architecture-specific findings.

- While cross-painting itself is not new (RoVi-Aug, Mirage), the contribution of scaling it to many-to-many augmentation and systematically studying the scaling effect on generalization is novel. The three AugE-Toolkit improvements (mask fusion, automatic base positioning, parallel deployment) are practical engineering contributions that enable this scaling.

**Weaknesses:**
- Physical experiments use only 10 trials per task per embodiment (Figure 5, line 424). The claimed 24% (OpenVLA) and 45% (π₀) improvements may not be statistically significant at conventional thresholds.
- All conditions train for the same number of steps, but N×Aug+Source has N× more unique trajectories than 1×Aug+Source, meaning each trajectory is seen fewer times. Part of the scaling gain could stem from a regularization effect (reduced overfitting) rather than embodiment diversity per se. Reporting training loss curves across conditions would help assess this.
- For a paper arguing that scaling augmentation matters, the absence of a performance-vs-number-of-embodiments curve (1, 2, 3, ..., N) is a notable gap. Only 0×, 1×, and N× are evaluated, which cannot reveal whether gains are linear, logarithmic, or exhibit diminishing returns. This limits the paper's utility as a scaling study.
- The method is fundamentally incompatible with dual-view (third-person + wrist) training, which is the standard setup for precision manipulation. Cross-painting can only augment third-person views where the robot body is visible; it cannot augment wrist-camera observations where the gripper (the key visual difference across embodiments) dominates the frame. All experiments use third-person observation exclusively (sim: Sec A.4; real: line 1042-1043). The wrist-view comparison (Sec A.6, Figure 11) only tests wrist-view *alone* as a strawman, sidestepping this fundamental issue. Since wrist cameras are widely recognized as essential for precision manipulation, this is an architectural limitation that significantly constrains the method's practical applicability, yet it is never discussed.
- The core augmentation approach (cross-painting via simulation rendering) is inherited from prior work (RoVi-Aug, Mirage). The improvements are practical but incremental. The paper's novelty lies primarily in the scaling study and dataset, not in the augmentation method itself. The paper would benefit from positioning itself more clearly as a dataset/benchmark contribution.

---

> ### Author Rebuttal · Authors · 2026-03-31
>
> #### **Q1: In the robustness experiments, why not compare against standard visual augmentations such as color jitter**
>
> We did not include color jitter in the paper because our goal was to isolate the effect of robot augmentation from basic image augmentation. This lets us show that robustness to lighting shifts and occlusions is not simply due to training on those perturbations, but emerges from embodiment diversity.
>
> However, we agree with your suggestion. We trained 0x Aug + Source policies with crop randomization, Gaussian noise, and color jitter, and compared them with robot-augmented policies.
>
> The results are shown in this plot: __https://pasteboard.co/9eZ0H5yEsZmZ.png__.
>
> Averaged over 50 rollouts across Square, Stack, and Two Piece Assembly, basic image augmentation improves over no augmentation, but robot augmentation provides a larger average benefit.
>
> #### **Q2: Suggestion: disentangling trajectory diversity from embodiment diversity.**
>
> Thank you for this suggestion. We’d like to clarify that in our current setup, we do not augment trajectories, only robot visuals. Each augmented sample replays the same source trajectory with a different robot embodiment; we do not use motion planning to generate different paths. Thus, policies are trained on the same set of underlying trajectories, with robot-augmented policies differing mainly in embodiment diversity rather than trajectory volume.
>
> Your suggestion is promising, and we agree that combining motion-planning-based augmentation with robot augmentation would be an interesting future direction for studying trajectory and embodiment diversity jointly.
>
> #### **Q3: The “unseen” Robotiq++ embodiment is visually different but not geometrically novel. Does this overstate the generalization claim?**
>
> We acknowledge that Robotiq++ is primarily a visually novel embodiment rather than a new gripper geometry, and we’ll clarify this in Sec. 6.1. That said, our broader simulation study does already evaluate transfer across multiple robot/gripper embodiments, which provides evidence that the method is useful beyond visual changes. At the same time, we agree that evaluating on a gripper with significantly different geometry would be an important extension and would further strengthen the generalization claim. We will clarify this scope and limitation more explicitly in the revision.
>
> #### **Q4: Cross-painting seems incompatible with dual-view setups**
>
> We would like to clarify that cross-painting is not limited to third-person views and can also be applied to wrist-camera viewpoints. We repeated the transfer and generalization experiments on three simulation tasks while augmenting both third-person and wrist views.
>
> This figure: __https://pasteboard.co/X0gLkZYDv6ph.png__ visualizes the cross-painting in the wrist camera frame.
>
> We compile the experiment results in this figure: __https://pasteboard.co/P3eCP5LP7VQB.png__.
>
> Results show similar scaling trends to the third-person-only setting. We will add these results and discussion in the revision.
>
> #### **Weakness 1: Physical experiments use only 10 trials per task per embodiment.**
>
> While we perform 10 trials per task per embodiment, this amounts to 80 trials per policy, and we report standard-error bars in Fig. 5. Averaged over 80 trials, OpenVLA improves from 48% without augmentation to 72% with Nx augmentation, and π₀ improves from 37% to 82%; both are statistically significant (p < 0.05).
>
> #### **Weakness 2: Part of the scaling gain could stem from a regularization effect rather than embodiment diversity per se.**
>
> Since our method is a form of data augmentation, some implicit regularization is expected. However, we believe this view is incomplete: in the generalization setting, the method still works even without target-robot data. This suggests the added data are not merely regularizing training, but also provide useful task-relevant supervision.
>
> We also examined training loss curves, see Fig: __https://pasteboard.co/88687YSuZWeP.png__, which appear broadly similar across conditions. This makes it less likely that the performance gains are due purely to reduced overfitting. The current evidence is consistent with both mechanisms, while the unseen-robot result suggests the effect is not *only* generic regularization.
>
> #### **Weakness 3: The absence of a performance-vs-number-of-embodiments curve (1, 2, 3, ..., N) is a notable gap**
>
> We appreciate this suggestion. Adding all intermediate datapoints is a bit too computationally expensive, since it would require training and evaluating many combinations of robot subsets and tasks. However, we have added 1/2/3/4x curves for the generalization setting:
>
> - Failure rate vs # Robots: __https://pasteboard.co/Mb4CASw7NGLZ.png__
>
> - Failure rate vs # Robots in Log Scale: __https://pasteboard.co/K7iALMqw9dev.png__
>
> These results suggest that generalization error on unseen robots decreases approximately logarithmically with respect to the number of training robots.

---

> > ### Author Rebuttal · Reviewer_xhs2 · 2026-04-01
> >
> > I appreciate the authors' rebuttal and the new experiments.
> > It's good to see that embodiment augmentation outperforms standard visual augmentation; the results could be a good addition to the paper.
> > - Q2 clarification. I think we're talking past each other. I wasn't suggesting trajectory augmentation via motion planning. What I meant: take 200 source Franka trajectories, then vary two axes, where trajectories used {50, 100, 200} and augmented embodiments {1, 2, 4, N}. A 2D grid like this would show whether gains come more from trajectory count or embodiment count, and how they interact. The current 0x/1x/Nx setup can't separate these. Even a small grid on one sim task would help practitioners decide how to allocate augmentation budget. But it's understandable if this is not feasible given the rebuttal timeline and it is already a good paper. Just thought we could draw some more insights from such an experiment.
> > - W5 (dual-view). The wrist-view cross-painting examples and dual-view results are a useful addition. The similar scaling trends do address my original concern about architectural incompatibility. However, the examples only show open Franka grippers swapped to other parallel-jaw grippers. Wrist-view inpainting is harder in practice: the gripper fills more of the frame, different geometries create different occlusion patterns, and the contact region between gripper and object is exactly where cross-painting is least reliable but most task-relevant. I'd suggest acknowledging these limitations in revision rather than treating wrist-view augmentation as fully solved.
> > - W3 (Robotiq++). Thanks for the honest acknowledgment that this is a visual change, not a geometric one. Please scope the generalization claims accordingly.
> > ---
> > [update] Thank you for the thorough follow-up experiment about embodiment augmentation vs trajectory augmentation, which is exactly what I was looking for. It directly addresses my earlier suggestion and provides actionable insights. The finding that embodiment diversity can be more effective than additional trajectories for generalization is a valuable practical takeaway that strengthens the paper's contribution as a scaling study.
> >
> > All of my concerns have been satisfactorily addressed. I am raising my score from 4 to 5 (accept).

---

> > > ### Author Response · Authors · 2026-04-07
> > >
> > > Thank you for the clarification, and we apologize for the earlier misunderstanding.
> > >
> > > **Regarding Q2,** we have added the experiment you suggested. We conducted experiments across varying degrees of robot augmentation using 50 and 100 source robot trajectories alongside our 200 source robot trajectory results, and analyzed the resulting grids over trajectory count × robot augmentation.
> > >
> > > We evaluated this on two tasks (Lift and Square), with performance averaged over all evaluation target robots. The resulting heatmaps and line plots are shown here:
> > >
> > > - Heatmap: https://pasteboard.co/T75X7KxfkOgf.png
> > > - Line plots: https://pasteboard.co/mINHkV91HgRB.png
> > >
> > > From these results, we observe that:
> > > - Robot augmentation improves performance across all trajectory budgets (50, 100, and 200).
> > > - Increasing the number of source trajectories improves performance across all robot augmentation levels.
> > > - The interaction between these two factors depends on task difficulty. For easier tasks such as Lift, gains begin to saturate near the performance ceiling in the Transfer setting; for example, N-way robot augmentation already reaches 0.91 with only 50 demonstrations, leaving limited room for further improvement.
> > > - For Lift in the Generalization setting and Square in the Transfer setting, performance scales roughly linearly with the number of demonstrations.
> > > - For the harder Square task in the Generalization setting, we observe a positive interaction between the two factors: increasing from 50→200 demonstrations yields a gain of only +0.04 with no augmentation, but +0.21 at $(N-1)$× augmentation with 50 demonstrations.
> > >
> > > A practical takeaway from these results is that, for generalization, embodiment diversity appears particularly valuable: increasing the number of embodiments can be more effective than allocating the same budget to additional trajectories from fewer embodiments. We appreciate this suggestion, as it helps clarify the relative roles of trajectory count and embodiment diversity, and we will include this analysis in the revision. We hope this addresses your concern more directly.
> > >
> > > **Regarding W5 (dual-view),** we agree with your suggestion. Wrist-view robot augmentation is feasible, but it is also more challenging to perform reliably at high quality. While our current results show that robot augmentation works well in relatively simple settings involving 2- or 3-jaw grippers, we have not yet studied more difficult cases in depth, such as heavier occlusion, substantially different camera viewpoints, or more complex gripper geometries. We will revise the paper to explicitly acknowledge these limitations.
> > >
> > > **Regarding W3 (Robotiq++),** thank you for this suggestion. We agree that this modification reflects a visual change rather than a geometric one, and we will scope the corresponding generalization claims more carefully in the revision.
> > >
> > > We appreciate these follow-up questions and suggestions. They have helped us improve both the analysis and the framing, and we hope the added experiments and clarifications address your concerns.

---

### Official Review · Reviewer_VkNg · 2026-03-15

**Soundness:** 3
**Presentation:** 3
**Significance:** 3
**Originality:** 3
**Overall Recommendation:** 5
**Confidence:** 4

**Summary:**

This paper presents OXE-AugE, a method and dataset from augmenting Open X-Embodiment into different robot embodiments with masking, inpainting, and sim. Experiments in real and sim show that scaling embodiment diversity in training improves robustness (eval performance in the same embodiment), transfer (eval performance in low-resource embodiments), and generalization (embodiments unseen during augmentation). The key takeaways are: embodiment augmentation improves robustness on the same robot compared to the visual input augmentation only baseline; embodiment augmentations help with transfer, with most of the gain coming from augmenting into the desired inference embodiment; embodiment augmentations help with generalization, with more embodiments performing the best.

**Compliance With Llm Reviewing Policy:**

Affirmed.

**Final Justification:**

See details above and rebuttal acknowledgement. Overall I think it's a decent paper and I maintain my accept recommendation.

**Key Questions For Authors:**

- How robust is the camera alignment / FoV search?
- In the physical experiments, what is the distribution of object initial positions for the evaluations?
- In A.7, could you clarify a bit why it would be necessary for the policy to take the predicted state instead of the ground-truth states for transfer?

**Limitations:**

Yes.

**Strengths And Weaknesses:**

- The paper is well-written with clear takeaways.
- The augmentation pipeline with inpainting + simulation replay is scalable to various behavioral cloning datasets.
- Experiments in real and sim provides strong evidence that the proposed augmentation pipeline improves performance for same-embodiment robustness, and cross-embodiment transfer and generalization.
- The augmentation assumes that the robot embodiments are similar enough that EEF tracking in sim + rendering overlay results in similarly valid trajectories, which can be violated by foreground objects / sufficiently large robot embodiment differences. The authors also mention this in the limitations.

---

> ### Author Rebuttal · Authors · 2026-03-31
>
> Thank you for your positive review and for recognizing the paper’s key takeaway that embodiment augmentation improves robustness, transfer, and generalization.
>
> #### **Q1: How robust is the camera alignment / FoV search?**
>
> Great question. We apply camera alignment and FoV search by maximizing the IoU between the simulation and the learned SAM2 mask using a discrete search, as explained in Appendix A.3.1. To analyze its robustness, we have plotted the IoU before and after this procedure on random samples of OXE-AugE. Please see this [histogram: https://pasteboard.co/RESK1lOLHQAl.png](https://pasteboard.co/RESK1lOLHQAl.png). Orange is the IoU before fusion, and blue is the IoU after fusion.
>
> Here is the summary statistics of the distributions:
> | Metric | IoU Before Fusion | IoU After Fusion |
> |---|---:|---:|
> | Mean IoU | 0.6314 | 0.8137 |
> | Std. Dev. | 0.0962 | 0.0595 |
>
> We note that the numbers are not very close to 1 because both SAM2 masks and simulation masks are imperfect (for example, SAM2 masks may over- or under-segment, while the simulation mask may be slightly different from the real robot). However, from this plot, we can see that after the alignment step, most of the frames have >0.7 IoU alignment and empirically we also find that >0.7 IoU aligns well visually. All the metadata is included in the augmented trajectories in OXE-AugE, and one can easily filter out outlier trajectories or frames for very low IoUs (eg. <0.6). We will add this analysis to our revision.
>
>
> #### **Q2: In the physical experiments, what is the distribution of object initial positions for the evaluations?**
>
> For each task in the physical experiments, we first uniformly randomly sample 10 paired positions for the object and target place locations from the workspace of the robot on the table (a roughly 70 cm x 50 cm region). Then for all policies and grippers, we use the same pre-selected positions for evaluation for fairness and to reduce result variance. We will clarify this protocol in the revision.
>
> #### **Q3: In A.7, could you clarify a bit why it would be necessary for the policy to take the predicted state instead of the ground-truth states for transfer?**
>
> Since OXE-AugE consists of many source datasets and robot embodiments, and each robot may have different dynamics (eg. gains) and control strategies (eg. velocity vs position) or even different action spaces (eg. joint vs Cartesian), we choose delta end-effector position as the common action space for the policy output instead of naively using each raw datasets’ original action. In other words, the policy predicts the actual delta state to be achieved: $s_{t+1} - s_t$.
>
> In deployment, our robot has yet another set of dynamics. This means that at state $s_t$, even if the policy predicts the desired delta end-effector movement to be $s_{t+1} - s_t$, the robot may not be at $s_{t+1}$ due to latency and compliance in control. If we feed in the actual achieved state $s_{t+1}’$ as the input to the policy at the next timestep, it may be slightly OOD and the undershooting or overshooting could accumulate and lead to inaccurate tracking. On the other hand, if we feed in $s_{t+1}$, the policy would predict $s_{t+2} - s_{t+1}$, and target goal position sent to the robot controller would be $s_{t+2}$, and even if the robot is not currently at $s_{t+1}$, it will aim to go to $s_{t+2}$ at the next timestep, thus mitigating the control errors and preventing it from accumulation.
>
> Above is an intuitive explanation of the idea behind action integration commonly used in control. In practice, the policy predicts an action chunk at each timestep instead of just one action, and we follow Policy-Level Action Integrator (PLAI) explained in Tang et al.$^*$ and find it effective. We will clarify this more explicitly in A.7.
>
> $^*$Tang, B., Lin, M. A., Akinola, I., Handa, A., Sukhatme, G. S., Ramos, F., ... & Narang, Y. Industreal: Transferring contact-rich assembly tasks from simulation to reality. RSS 2023.

---

> > ### Author Rebuttal · Reviewer_VkNg · 2026-04-05
> >
> > Thank you for the clarifications and additional information. I think this is a solid paper and recommend its acceptance.

---

### Decision · Program_Chairs · 2026-04-30

**Decision:**

Accept (spotlight)

**Comment:**

Reviewers were uniformly positive with no major score disagreements, agreeing on the main takeaway that diverse multi-robot augmentation fundamentally improves policy generalization to novel unseen embodiments. The release of the massive dataset and the practical pipeline that cleverly bypasses strict camera calibration requirements are highly useful. Some initial concerns were understandably raised regarding the restriction to third-person views and the relative simplicity of the manipulation tasks evaluated. However the authors delivered a highly effective rebuttal that compressed numerous complex new evaluations into the short window, notably adding wrist-camera experiments and a very insightful grid analysis disentangling trajectory volume from embodiment count. These critical additions fully resolved the main criticisms and solidifed the paper as a technically sound and highly useful resource. I recommend this paper for acceptance.